# Role of vertical and horizontal mixing in the tape recorder signal near the tropical tropopause

Anne A. Glanville[1] and Thomas Birner[1]

[1]Department of Atmospheric Science, Colorado State University, Fort Collins, CO, USA

*Correspondence to:* T. Birner (thomas.birner@colostate.edu)

**Abstract.** Nearly all air enters the stratosphere through the tropical tropopause layer (TTL). The TTL therefore exerts a control on stratospheric chemistry and climate. The hemispheric meridional overturning (Brewer-Dobson) circulation spreads this TTL influence upward and poleward. Stratospheric water vapor concentrations are set near the tropical tropopause and are nearly conserved in the lowermost stratosphere. The resulting upward propagating tracer transport signal of seasonally varying entry concentrations is known as the tape recorder signal. Here, we study the roles of vertical and horizontal mixing in shaping the tape recorder signal in the tropical lowermost stratosphere, focusing on the 80 hPa level. We analyze the tape recorder signal using data from satellite observations, a reanalysis, and a chemistry-climate model (CCM). Modifying past methods, we are able to capture the seasonal cycle of effective vertical transport velocity in the tropical lowermost stratosphere. Effective vertical transport velocities are found to be multiple times stronger than residual vertical velocities for the reanalysis and the CCM. We also study the tape recorder signal in an idealized one-dimensional transport model. By performing a parameter-sweep we test a range of different strengths of transport contributions by vertical advection, vertical mixing, and horizontal mixing. Introducing seasonality in the transport strengths we find that the most successful simulation of the observed tape recorder signal requires multiple times stronger vertical mixing at 80 hPa compared to previous estimates in the literature. Vertical mixing is especially important during boreal summer when vertical advection is weak. Simulating the reanalysis tape recorder requires excessive amounts of vertical mixing compared to observations but also to the CCM, which hints at the role of spurious dispersion due to data assimilation. Contrasting the results between pressure and isentropic coordinates allows further insights into quasi-adiabatic vertical mixing, e.g. associated with overshooting convection or breaking gravity waves. Horizontal mixing, which takes place primarily along isentropes due to Rossby wave breaking, is captured more consistently in isentropic coordinates. Overall our study emphasizes the role of vertical mixing in lowermost tropical stratospheric transport, which appears to be as important as vertical advection by the residual mass circulation. This questions the perception of the 'tape recorder' as a manifestation of slow upward transport as opposed to a phenomenon influenced by quick and intense transport through mixing, at least near the tape head. However, due to limitations of the observational data set used and the simplicity of the applied transport model, further work is required to more clearly specify the role of vertical mixing in lowermost stratospheric transport in the tropics.

# 1 Background

Water vapor accounts for less than 0.001% of stratospheric air, but as a radiatively active tracer it plays a major role in shaping its climate. Even surface temperature can be radiatively affected by changes in stratospheric water vapor on decadal time scales (Solomon et al., 2010) and the near-surface circulation may respond to these changes through downward coupling (Maycock et al., 2013).

Most water vapor enters the stratosphere through an interface known as the tropical tropopause layer (TTL) from where it spreads upward and poleward along the Brewer-Dobson circulation (BDC) (Brewer, 1949; Butchart, 2014). The extremely low temperatures in the TTL cause dehydration by freeze-drying and therefore determine the amount of water vapor that enters the stratosphere (Fueglistaler et al., 2009). Water vapor above the TTL behaves nearly like a passive tracer. Concentrations are stamped at the base of the stratosphere by the annual cycle in tropical tropopause temperature and moved upward by the BDC, creating the so-called tape recorder signal in the tropical lower stratosphere (Mote et al., 1996). By exploiting water vapor as a tracer for lower stratospheric transport, we can investigate the speed of BDC upwelling and the relative importance of mixing versus advection.

The TTL is a transition region between convective outflow in the upper troposphere ∼200 hPa and the base of the deep branch of the BDC ∼70 hPa. This region features a mix of tropospheric and stratospheric properties and is controlled by complex interactions between dynamics, clear-sky radiation and its coupling to transport of radiatively active tracers, as well as cloud-radiative effects and cloud microphysics. Dynamical control acts on a vast range of scales, including planetary-scale circulations, equatorial waves, and convection (Randel and Jensen, 2014). The BDC is a measure of aggregated transport on all spatial and temporal scales (Butchart, 2014) and may provide insight into different transport contributions at and just above the TTL. There are currently no direct measurements of the magnitude or variability of tropical upwelling near the tropical tropopause (e.g. Abalos et al., 2013).

The tape recorder signal emerges when plotting the time-height sections of zonally-averaged water vapor in the tropical lower stratosphere. Figure 1 shows the tape recorder signal obtained from Microwave Limb Sounder (MLS) measurements. Although the transport through the TTL and lower stratosphere is strongly guided by slow upward advection due to the residual mean meridional mass circulation (e.g. Holton et al., 1995), recent studies have emphasized the importance of vertical and horizontal mixing on the overall transport (Flannaghan and Fueglistaler, 2014; Konopka et al., 2007; Ploeger et al., 2011; Sargent et al., 2014), especially near the tape head (the tropical tropopause).

At the tape head, water vapor has a strong seasonal cycle with anomalously high values during boreal summer and anomalously low values during boreal winter. This is a direct result of the seasonal cycle in the temperature of the cold point tropopause (CPT) – anomalously warm during boreal summer and anomalously cold during boreal winter – which affects the water vapor content of the air through the process of freeze-drying (dehydration). As a result of tropical upwelling there is a phase lag between the signal at the base versus the signal at higher altitudes. Interannual variability associated with the quasi-biennial oscillation (QBO) and the El Niño Southern oscillation (ENSO) also impact water vapor transport through the TTL and lower stratosphere (e.g. Davis et al., 2013).

Chemistry-climate models (CCMs) show a large (10K) spread in annual mean CPT temperatures and these discrepancies have been associated with their differing transport characteristics (Gettelman et al. (2009) and Eyring et al. (2010)) and even details of the numerical schemes (Hardiman et al., 2015). As mentioned above, these temperatures control the amount of water vapor entering the stratosphere with consequences for the models' radiation budget. Improved transport characteristics on various scales might help to narrow the models' CPT temperature spread. More accurate modeling of TTL processes is expected to result in improved calculations of the global radiation balance, which is important for future climate predictions. But accurate simulations of TTL transport require improved understanding of the dynamics in this region.

Horizontal mixing and slow upwelling near the tropical tropopause are closely related because both are driven by Rossby wave breaking occurring between the tropics and extratropics. On the other hand, vertical mixing in the TTL and lowermost stratosphere may be directly or indirectly associated with tropical deep convection. Overshooting convection directly leads to mixing but is limited by the depth of the overshoots. Gravity waves and other equatorial waves associated with deep convective clouds can propagate vertically into the tropical stratosphere (Kiladis et al., 2009). When these waves dissipate they may cause vertical mixing, which is then indirectly associated with the convection. Deep convection also influences water vapor concentrations in the TTL either directly through lofting of ice with subsequent sublimation (e.g. Kuepper et al., 2004), or indirectly through dehydration associated with the large-scale tropopause-level cold response to upper-tropospheric heating (e.g. Johnson and Kriete, 1982; Holloway and Neelin, 2007; Paulik and Birner, 2012).

The purpose of this study is to quantify the individual contributions to total transport of water vapor above the tropical tropopause in hopes to improve our understanding of the multi-scale nature of the dynamics in this region—from quick, small-scale vertical mixing to slow, large-scale residual vertical advection. Part of this study takes advantage of an isentropic coordinate (i.e. quasi-Lagrangian) framework to visualize transport. Horizontal mixing between the tropics and mid-latitudes is quasi-adiabatic and therefore best described in isentropic coordinates (e.g. Konopka et al., 2007; Ploeger et al., 2011). Vertical transport in isentropic coordinates is by definition directly related to diabatic heating. A component of vertical mixing, e.g. due to overshooting convection or breaking small-scale gravity waves, may be assumed to take place quasi-adiabatically[1] and will therefore leave different signatures in isentropic versus pressure or height coordinates.

The paper is organized as follows. Section two and three describe the data and methods used in this study, respectively. Sections four and five present the results in pressure and isentropic coordinates, respectively. Our results are discussed in section six.

## 2   Data

Water vapor is a quasi-conserved tracer in the TTL and lower stratosphere and therefore offers insights into total transport. The slope of water vapor isolines in a time-height plot is a measure of the effective upward speed of the BDC. The Microwave

---

[1]In the region of interest – the tropopause and lower stratosphere – the dominant contribution to diabatic heating is due to radiation (possibly including cloud-radiative effects). We assume that the time scales for mixing by small-scale processes are generally much shorter than the relevant radiative time-scales. The mixing may then be assumed to take place quasi-adiabatically.

Limb Sounder (MLS) aboard the NASA Aura satellite, launched in 2004, offers daily coverage with ~3.5 km vertical resolution within the TTL and nearly global horizontal coverage. These measurements are reliable in the presence of aerosol or cirrus clouds. We use MLS version 3.3 (v3.3) data obtained from the Aura website (http://mls.jpl.nasa.gov/index-eos-mls.php) following the data quality screening given in the MLS data quality document (Livesey et al., 2007). MLS' vertical resolution results in relatively coarse sampling of the tropical lowermost stratosphere (e.g. the averaging kernel for the $\sim 80$ hPa level includes a $\sim 20\%$ contribution from 100 hPa). At a number of places we therefore also include results using the older HALOE data set (Russell et al., 1993), which has doubled vertical resolution compared to MLS.

We focus on the inner tropics by employing a $10°$S–$10°$N latitude average, which ensures sufficient sampling and covers the latitudinal variations in the location of maximum upwelling. Tests with a slightly bigger latitude range of $15°$S–$15°$N resulted in only minor quantitative modifications of our results.

To enhance our understanding of transport processes and to test our methods we also employ the European Centre for Medium-Range Weather Forecasts (ECMWF) Interim Reanalysis (ERA-i) on a Gaussian grid at T255 spectral resolution ($\sim$80 km or $\sim$0.7°) on the 60 vertical model levels. The available data spans from 1 January 1979 to present with 6-hourly temporal resolution, but we focus on the time frame that overlaps with MLS. Tropical stratospheric transport in ECMWF's previous reanalysis system, ERA-40, was twice as fast as that in ERA-i (Dee et al., 2011). For example, the moist and dry signals of ERA-40's tape recorder signal reached 30 hPa only about three months after leaving the 100 hPa level. In ERA-i, the transport between those surfaces takes six months, closer to reality. Nonetheless, this is still at least twice as fast compared to MLS observations as can be seen in Figure 2 where dotted lines roughly indicate the evolution of the dry signal for each dataset (cf. Jiang et al., 2015). ERA-i does not assimilate stratospheric water vapor. However, given how strong of a function of the cold point temperature it is, and given that temperatures are assimilated, ERA-i's stratospheric water vapor should not be considered to be unconstrained. In fact, Fig. 2 shows that apart from the tape recorder seasonality (i.e. transport strength), ERA-i and MLS agree quite well in the stratosphere (in terms of overall absolute values).

To better understand the influence of data assimilation on transport in ERA-i, we also analyze the tape recorder in the Goddard Earth Observing System (GEOS) Chemistry Climate Model (CCM) without data assimilation. Schoeberl et al. (2008b) also compared effective vertical transport velocities between MLS and the GEOS-CCM, so using the same model eases comparison to previous work. The GEOS CCM combines atmospheric chemistry and transport modules with NASA's GEOS circulation model. The GEOS CCM took part in the Chemistry Climate Model Validation 2 activity (CCMVal-2) which included other stratosphere-resolving, interactive-chemistry models performing historical (REF-B1) and future (REF-B2) runs. The historical runs do not overlap with the MLS period. We therefore use the REF-B2 run to analyze the same time period as available from MLS. Compared to all other models in CCMVal-2, GEOS CCM was found to produce one of the best simulations of mean age of air, a measure of the BDC speed. Eyring et al. (2010) found the CCM's residual circulation in the lower stratosphere to be somewhat slower than what is implied through its tape recorder, however our improved effective velocity method shows it to be comparable in the annual mean. We will show that the separation between GEOS CCM and ERA-i residual circulations is much smaller than the separation between their effective velocities, implying an impact on transport by data assimilation.

## 3 Methods

We use two methods to study transport in the tropical lowermost stratosphere. First we analyze the tape recorder signal to estimate the effective vertical transport velocity as a measure of BDC tropical upwelling just above the tropical tropopause, expanding on previous work in the literature. Second we study the relative roles of residual vertical advection, vertical, and horizontal mixing using a one-dimension advection-diffusion-dilution model similar to that in Mote et al. (1998). We also use this simple, idealized model to test the efficacy of the first method.

For altitudes higher than 21 km ($\sim 40$ hPa), methane oxidation acts as a source for water vapor and upon reaching 25 km ($\sim 25$ hPa), about 0.25-0.5 ppmv is added to the signal (e.g. Mote et al., 1998; Schoeberl et al., 2012). Here, we focus on the lowermost stratosphere where this effect can be largely neglected.

### 3.1 Effective vertical transport velocity

We follow Schoeberl et al. (2008b) and use phase-lagged correlations between adjacent levels of the tape recorder signal to estimate an effective vertical transport velocity, a method previously introduced by Niwano et al. (2003) and recently used in modified form by Minschwaner et al. (2016). The earlier studies used large sample sizes ($\sim 1$ year) to compute the correlations. These sample sizes tend to highlight interannual variability (such as due to the QBO) over seasonal variability. Here, we modify this method to parse out shorter-duration variability. First, we obtain correlation coefficients between daily data at consecutive levels. The data at the higher level are then shifted in 1-day increments up to 14 months to find the largest correlation coefficient. Strong correlation between the data at the lower level and the shifted data at the higher level is assumed to follow the tape recorder. The effective transport vertical velocity, assigned to midpoints between levels and time steps, is simply the distance between the levels divided by the time-shift associated with the largest correlation coefficient. We consider effective transport velocities in both pressure and isentropic coordinates. Vertical velocities in pressure coordinates will be presented as log-pressure velocities to give the more often used unit of mm s$^{-1}$, using a constant scale height of 7 km[2] .

Instead of using a large ($\sim 365$ days) sample size for computing the correlation coefficients (Schoeberl et al., 2008b), we have found a sample size of $\sim 180$ days days capable of parsing out the seasonal cycle of effective transport velocity. Further, unlike Schoeberl et al. (2008b), we retain high correlations that occur at lags of less than one month. However, lags of less than seven days are omitted because they produce unrealistic and temporally unvarying speeds with low correlations. Our modified phase-lagged correlation method was tested on a synthetic tape recorder signal with varying advection scenarios. Results show that the method is more likely to underestimate by 0.05 hPa day$^{-1}$ below 60 hPa and more likely to overestimate by 0.05 hPa day$^{-1}$ above 60 hPa. Small vertical velocities in the middle stratosphere and rapid water vapor changes in time are not fully identified (e.g., in May when the signal goes from dry to moist). Overall, the method appears to successfully capture the seasonality and magnitude of the transport.

---

[2]A more appropriate scale height for the tropical lowermost stratosphere would be 6 km ($H = RT_0/g$, where $R$ is the gas constant for dry air, $T_0$ is a reference temperature, $g$ is the acceleration due to gravity). However, we opt for 7 km as this is the most commonly used scale height in the expression for log-pressure coordinates (e.g. Andrews et al., 1987)

We emphasize that this lag-correlation method based on the observed tape recorder signal results in an effective (vertical) transport velocity. When mixing has negligible influence on the signal this velocity may be assumed to be approximately equal to the residual vertical velocity (Schoeberl et al., 2008b). However, especially in the lowermost tropical stratosphere the effects of horizontal and vertical mixing may be significant. Vertical mixing will cause the signal to spread between two levels while

reducing the time lag for maximum correlation and therefore increase the inferred velocity. The influence of horizontal mixing is to dilute the tape recorder signal (Mote et al., 1998), but depends on the horizontal background structure that is seasonally varying.

## 3.2   One-dimensional model

Estimates of the effects of vertical and horizontal mixing on the tape recorder signal may be obtained by simulating this signal

with a one-dimensional transport model:

$$\partial_t \overline{\chi} = -\overline{\omega}^* \partial_p \overline{\chi} + \partial_p(K_p \partial_p \overline{\chi}) - \alpha_p(\overline{\chi} - \overline{\chi}_{ML}) + S. \tag{1}$$

Here, $\overline{\chi}$ is the water vapor mixing ratio, $\overline{\omega}^*$ is the residual vertical velocity, $K_p$ is the vertical diffusivity in pressure coordinates, $\alpha_p$ is the horizontal dilution rate in pressure coordinates, $\overline{\chi}_{ML}$ is the mid-latitude (here, ) reference value of $\overline{\chi}$, and overbars represent the zonal mean. $\overline{\chi}_{ML}$ is obtained from the actual (seasonally varying) MLS, HALOE, or ERA-i data, averaged over

30° to 60° latitude. We have tested an alternative latitude range for $\overline{\chi}_{ML}$ of 15° to 45° and found no significant changes to our results. $S$ is a chemical source-sink term. We set $S = 0$ because we are only interested in the tape recorder below the level of methane oxidation, which becomes important above ∼40 hPa (Dessler et al., 1994). This also neglects cloud formation or evaporation just above the tropopause, as well as a potential contribution due to dehydration at the local cold point tropopause. We will discuss this potential drawback in detail in section 6. Our model is similar to the one used in Mote et al. (1998) except

that it uses pressure coordinates (we also use a potential temperature coordinate version, see below).

Mote et al. (1998) solved for annual mean parameters by defining the tape recorder as a wave solution and inverse-solving for advection, diffusion (vertical mixing), and dilution (horizontal mixing). Although they tested their model on synthetic data, the solutions from this approach are restricted because they rely on the tape recorder fitting a perfect wave at each level, which may be problematic in the presence of mixing. The most severe restriction, however, comes from using an annual mean

value for the residual vertical velocity. It is by now well established that the strength of residual tropical upwelling undergoes a significant seasonal cycle, with smaller values in boreal summer (e.g. Butchart, 2014). Vertical transport due to residual vertical advection alone slows down significantly during boreal summer, enhancing the relative importance of mixing to total transport particularly in this season. Assuming annual mean values for the transport parameters essentially underestimates the contribution due to mixing. We therefore introduce seasonality in these parameters by prescribing reductions and enhancements

of 50% over the course of the seasonal cycle. The 50%-value results in realistic variations for vertical advection, corresponding to estimates in the literature (e.g. Rosenlof, 1995; Abalos et al., 2013).

Seasonal variations in the mixing strengths are less well constrained by past studies; for these we use the same value of 50%. We tested other seasonal cycle amplitudes for the MLS data set in pressure coordinates by keeping a 50% amplitude in

two of the transport parameters while varying the third (ranging from 0% to 100%). Overall, we find that the seasonality in the transport parameters has little influence on the tape recorder as long as the amplitude remains below 75% for the vertical transport parameters. Small improvements in the simulated tape recorder signal can be obtained by slight adjustments of the seasonal cycle amplitude in vertical advection or vertical mixing. In comparison, the tape recorder signal is hardly affected by changes in the seasonal cycle amplitude of the horizontal mixing strength. Given the mostly insignificant changes in the simulated tape recorder signal for modest changes in seasonal cycle amplitudes, we stick to the 50% values in this study for simplicity.

We further remove the perfect wave restriction by running a parameter sweep with varying strengths of each transport. Control values for the annual mean solutions (denoted by subscripts 'ctrl') are taken to be the solutions obtained by Mote et al. (1998), including their vertical structure. Transport strengths are varied from 0 to 10 times their control value. Apart from these modifications, our model carries the same assumptions as discussed by Mote et al. (1998). It assumes that tropical air is horizontally well-mixed within the latitude bounds (here, 10°S to 10°N) and is notably different, though not completely isolated, from mid-latitude air. The vertical eddy water vapor flux in the full water vapor budget can be represented as instantaneous diffusion acting on the vertical gradient of water vapor ($\overline{\omega'\chi'} \simeq -K_p \partial_p \overline{\chi}$, with $K$ a positive constant, further discussed below). Horizontal mixing by midlatitude air is modeled by a linear relaxation process (dilution) in which tropical air is relaxed towards $\overline{\chi}_{ML}$ with rate $\alpha_p$. This last assumption represents a crude approximation – horizontal mixing in the lowermost stratosphere is generally a more complex process (Konopka et al., 2009; Ploeger et al., 2011). Note that by taking $\overline{\chi}_{ML}$ from the actual seasonally varying data, we expect to better represent influences due to the monsoon circulations, e.g. during boreal summer (e.g. Gettelman et al., 2004).

We prescribe the seasonal cycle of advection ($\overline{\omega}^*$) as a sine wave that peaks during boreal winter (on January 1st) when the meridional circulation is strongest according to observations. Vertical diffusion ($K$) is prescribed with the same seasonality, i.e. strongest during boreal winter, consistent with the results in Flannaghan and Fueglistaler (2014) and when convective influence on the TTL is strongest (Fueglistaler et al., 2009). While observational estimates of vertical mixing and its seasonal cycle are sparse to nonexistent, this seasonal cycle can be considered to be a plausible first guess, and is found to not have a strong influence on our results (see below). The seasonal cycle of horizontal mixing $\alpha_p$ is opposite from that of vertical advection and vertical mixing. Horizontal mixing is prescribed to maximize during boreal summer (on July 1st) when the subtropical mixing barrier (jet) is relatively weak (Gettelman et al., 2011; Ploeger et al., 2012). We tested the model with seasonality in vertical advection only, which resulted in somewhat lower performance with respect to its ability to reproduce the observations, but the main qualitative features of our results to be presented in section 4 are not affected.

We also use the one-dimensional transport model in isentropic coordinates. This has the advantage that the representation of horizontal mixing becomes more realistic – this process is driven by Rossby wave breaking and takes place approximately along isentropes in the real atmosphere. Furthermore, vertical mixing is partially an adiabatic process (e.g. if driven by small-scale gravity wave breaking) and is therefore expected to contribute less to transport through isentropic surfaces. On the other hand, comparison to our data sets is more straightforward in pressure coordinates, so additional insight may be gained by

comparing the two coordinate systems. In isentropic coordinates the model may be written as

$$\partial_t \overline{\chi}^* = -\overline{Q}^* \partial_\theta \overline{\chi}^* + \overline{\sigma}^{-1} \partial_\theta(\overline{\sigma} K_\theta \partial_\theta \overline{\chi}^*) - \alpha_\theta(\overline{\chi}^* - \overline{\chi}^*_{\mathrm{ML}}) + S, \tag{2}$$

where $Q$ is the diabatic heating rate and $\sigma$ is isentropic (mass) density (also often referred to as thickness). Overbars with asterisks denote mass-weighted zonal averages (e.g. $\overline{\chi}^* \equiv \overline{\sigma\chi}/\overline{\sigma}$).

As measures of the model's performance in simulating the tape recorder we analyze the amplitude, phase, and annual mean of water vapor mixing ratio at 80 hPa and 400 K for each parameter combination. The phase is obtained using simple Fourier analysis, while the amplitude is obtained simply from the minimum and maximum values. We introduce a score (out of 100%, see equation below) that is a function of the multiplying factors $(a, b, c)$ on the control values of residual vertical velocity or diabatic heating rate, vertical diffusivity, and horizontal dilution rate. For example, in pressure coordinates the factors $(a, b, c)$

determine the values of $(\overline{\omega}^*, K_p, \alpha_p) = (a\overline{\omega}^*_{\mathrm{ctrl}}, bK_{p,\mathrm{ctrl}}, c\alpha_{p,\mathrm{ctrl}})$. Generally we find that the strengths of vertical advection and horizontal mixing are not independent and their variations result in similar structures (i.e., $a = c$ with the high-scoring combination). This is perhaps not surprising as both are a function of subtropical Rossby wave breaking (e.g. Garny et al., 2014). To highlight that typically $a = c$ we denote the combined effects of vertical advection and horizontal mixing by $G$, with the control value $G_{ctrl}$ for $a = c = 1$. There are rare cases where the original Mote et al. (1998) values for vertical advection

and horizontal mixing must be multiplied by different factors to create the highest score (i.e. where $a \neq c$). In these cases $G_{ctrl}$ represents $a$. For example, if an optimal solution requires $(a, b, c) = (1, 1, 3)$, then $1 \times G_{ctrl}$ corresponds to $a = 1$ and $c = 3$, while $2 \times G_{ctrl}$ corresponds to $a = 2$ and $c = 6$, and so on. These rare cases will be discussed separately.

The score as a function of $(a, b)$ (assuming $c = a$) is:

$$\mathrm{score}(a, b) = \frac{100}{1 + \frac{|A_s - A_r|}{|A_r|} + \frac{|\phi_s - \phi_r|}{|\phi_r|} + \frac{|\chi_s - \chi_r|}{|\chi_r|}}, \tag{3}$$

where $A$ is the amplitude, $\phi$ is the phase, and $\chi$ is the water vapor mixing ratio. Subscripts "$s$" and "$r$" refer to the synthetic and real tape recorder signals, respectively.

## 4   Results in pressure coordinates

### 4.1   Effective vertical transport velocity

Both MLS and HALOE show seasonal variations in effective vertical transport velocity in the TTL and lower stratosphere,

with stronger upwelling during boreal winter (Figure 3). Boreal winter upward transport is 2-3 times stronger than during summer in these data sets. Finer vertical resolution in HALOE results in more consistent vertical spreading of this seasonality up to $\sim 50$ hPa. The difference in the depth of the signal may be due to our method underestimating small speeds, which may be more pronounced in MLS with its coarser vertical resolution. Seasonal variations are qualitatively similar in ERA-i, but velocities are 2–4 times greater than in MLS and HALOE.

Figure 4 highlights that the inferred effective vertical transport velocity is not necessarily the same as the residual upward velocity. In ERA-i the effective vertical transport velocity is about 4 times larger than the residual vertical velocity at 80 hPa,

which points to the role of vertical and/or horizontal mixing in transport just above the tropical tropopause (and amplified dispersion due to data assimilation, see below). The MLS and HALOE derived transport velocities are of similar magnitude as the residual circulation velocity in ERA-i. Taking into account that ERA-i's residual vertical velocity seems biased high near the tropical tropopause (Abalos et al., 2015), this indicates that effective vertical transport is stronger than by the residual

circulation alone. The double-minimum structure between April–August in MLS effective transport velocity is likely the result of noisy data around the transition between the wet and the dry part of the signal.

    The residual vertical velocity is about 25% weaker in GEOS CCM (green lines in Fig. 4) than ERA-i, albeit with identical seasonality. Its effective vertical transport velocity, however, only shows very little seasonal variation and is significantly smaller than in ERA-i, in closer agreement with MLS and HALOE during boreal winter. The large difference between the

GEOS CCM and ERA-i inferred effective transport velocities (up to 4 times larger in ERA-i) suggests that excessive vertical dispersion due to data assimilation dominates in ERA-i. Mixing appears to have a stronger influence on transport in boreal summer in GEOS CCM (cf. difference between green dashed and full lines in Fig. 4).

## 4.2   One-dimensional transport modeling

Figure 5 shows that a range of combinations that slightly vary $G$ but more so $K$ result in high-scoring simulations of observed

water vapor at 80 hPa for both MLS and HALOE. Moderately high scores may be achieved by using the control value for vertical mixing ($K_{ctrl}$), but require increases in vertical advection and horizontal mixing by more than 50% of their control values. Using $G_{ctrl}$ on the other hand, a near-perfect score (near or above 90%) results from increasing $K$ by a factor of 4 for MLS or 5.5 for HALOE. The strength of vertical advection may be considered to be better constrained from past studies (e.g. Rosenlof, 1995; Plumb, 2002), while the strength of vertical mixing remains more ambiguous. Closer inspection of the

individual tape recorder characteristics shows that high-scoring (above 90%) simulations of its amplitude alone require at least $3 \times K_{ctrl}$. High-scoring simulations of only its phase, on the other hand, are more sensitive to the strength of vertical advection and to allowing transport seasonality (particularly vertical advection).

    Figure 6 compares the tape recorder signal between our simple model and the satellite observations using the transport combination $(1 \times G_{ctrl}, 4 \times K_{ctrl})$ for MLS and $(1 \times G_{ctrl}, 5.5 \times K_{ctrl})$ for HALOE (white stars in Figure 5). The time series

at 80 hPa (bottom panels) further shows that our parameter settings better capture the observed seasonal water vapor evolution than the Mote et al. (1998) control setting, although the seasonal cycle amplitude is still somewhat underestimated. Note that the Mote et al. (1998) solution much better captures the HALOE time series (on which it was based).

    Inspecting the individual transport contributions to the time tendency of water vapor for MLS (Figure 7) shows that vertical advection and vertical mixing play equally significant roles in forming the tape recorder signal at 80 hPa. Vertical mixing plays

a particularly large role during late summer / early fall. Transport contributions based on HALOE are very similar (see Fig. S2 in supplement).

    Horizontal mixing generally plays a small role, except during boreal spring. Note that the tendency due to horizontal mixing is a function of both the dilution rate $\alpha_p$ and the background meridional gradient ($\sim \overline{\chi} - \overline{\chi}_{ML}$). The latter becomes small during boreal summer when meridional mass transport maximizes and this explains why the tendency due to horizontal mixing

maximizes during boreal spring. This is consistent with the results in Ploeger et al. (2012): their Fig. 1a shows from trajectory calculations that the difference in water vapor mixing ratio between in-mixed extratropical air and tropical air is largest during boreal spring and becomes negligible during boreal summer. Nevertheless, the strongest signature in tropical mean water vapor will be found when the accumulated tendency is strongest: this happens during late boreal summer when the tendency (green line in Fig. 7) crosses through zero.

High-scoring simulations of the ERA-i tape recorder signal in pressure coordinates require much greater amounts of vertical mixing than for the MLS or HALOE observations (Figure 5). Based on all parameter combinations tested, vertical mixing needs to be at least an order of magnitude larger than the control values. We found that high-scoring solutions also require strongly enhanced horizontal mixing (multiple times its control value), whereas vertical advection may remain unchanged from its control value (small changes in it require large changes in both types of mixing to compensate). $G_{ctrl}$ therefore corresponds to $c = 3a$ in the transport combinations for ERA-i and the x-axis in Figure 5b only extends to $3G_{ctrl}$ because changes in transport strengths by more than one order of magnitude have not been examined.

The one-dimensional model results imply that eddy transport in the lowermost stratosphere is strongly enhanced in ERA-i compared to observations, especially in the vertical. Amplified vertical advection alone does not result in much improved tape recorder simulations. In fact, even reduced vertical advection may easily be compensated by slight further amplifications of vertical and horizontal mixing. The enhanced eddy mixing in ERA-i is likely a result of spurious dispersion due to data assimilation (Schoeberl et al., 2003), but could also result from diffusive numerical schemes. In the case of the GEOS CCM, a transport combination more similar to MLS or HALOE produces the highest scores – enhanced vertical mixing with vertical advection and horizontal mixing near their control values (not shown). This difference between the free-running model and the reanalyses further points to excessive dispersion due to data assimilation in ERA-i. We also note that our simulations of the GEOS CCM tape recorder signal are not very sensitive to changes in vertical mixing strength, which is likely due to its small vertical water vapor gradient so that vertical diffusion remains small.

## 5  Results in isentropic coordinates

### 5.1  Effective vertical transport velocity

In isentropic coordinates the effective vertical transport velocity corresponds to diabatic heating. Figure 8 shows this diabatic effective vertical transport velocity for MLS and ERA-i using the phase-lagged correlation method as before. Averages (zonally and in time) in isentropic coordinates are appropriately obtained by applying mass-weighting, which is implicit in pressure coordinates. The seasonal cycles of diabatic heating rates thus obtained are similar between MLS and ERA-i, with maxima in the lowermost stratosphere during boreal winter, as expected. Maximum diabatic heating from MLS is $\sim 1$ K/day, that from ERA-i is 4-5 times larger and located slightly higher (at 410 K versus 390 K for MLS, perhaps related to temperature differences in this region). The enhanced diabatic heating in ERA-i compared to MLS is consistent with Wright and Fueglistaler (2013) and Yang et al. (2010), who found longwave cloud radiative heating rates above 200 hPa to be larger in ERA-i compared to other reanalyses and a detailed radiative transfer model. Wright and Fueglistaler (2013) note that water vapor contents and therefore

treatment of convective anvil clouds in ERA-i could partially explain the anomalous heating rates. ERA-i has also been found to exhibit $\sim 40\%$ too large clear-sky radiative heating rates (Ploeger et al., 2012). However, the discrepancy between MLS and ERA-i is likely also due to excessive vertical and horizontal dispersion as discussed in the previous section.

The difference between the effective vertical transport velocities and the contributions due to vertical and horizontal mixing may be better understood by considering the zonal mean tracer evolution equation, which in pressure coordinates reads (written in Cartesian coordinates and neglecting sources and sinks for simplicity):

$$\partial_t \overline{\chi} + \overline{\omega} \partial_p \overline{\chi} + \overline{v} \partial_y \overline{\chi} = -\partial_y \overline{v' \chi'} - \partial_p \overline{\omega' \chi'}\,.$$

Here, $v$ is the meridional velocity and primes denote deviations from the zonal mean (denoted by overbars as before). The effective vertical transport velocity ($\omega_{\mathrm{eff}}$) results formally from setting:

$$\partial_t \overline{\chi} + \omega_{\mathrm{eff}} \partial_p \overline{\chi} = 0\,,$$

hence:

$$\omega_{\mathrm{eff}} = \overline{\omega} + (\partial_p \overline{\chi})^{-1} (\overline{v} \partial_y \overline{\chi} + \partial_y \overline{v' \chi'} + \partial_p \overline{\omega' \chi'})\,.$$

This shows how both, horizontal and vertical eddy fluxes ($\sim$ mixing) lead to differences between $\overline{\omega}$ and $\omega_{\mathrm{eff}}$ (note that in the residual, transformed-Eulerian, mean form, horizontal mixing is partially included in $\overline{\omega}^*$ – more precisely, the part that is aligned with the meridional eddy heat flux, Andrews et al. (1987); so in that form it is primarily the vertical mixing that creates differences between $\overline{\omega}$ and $\omega_{\mathrm{eff}}$). Horizontal advection may cause an additional difference but is generally small in the deep tropics.

In isentropic coordinates, the corresponding zonal mean tracer evolution equation reads:

$$\partial_t \overline{\chi}^* + \overline{Q}^* \partial_\theta \overline{\chi}^* + \overline{v}^* \partial_y \overline{\chi}^* = -\overline{\sigma}^{-1} \partial_y \overline{\hat{v} \sigma \hat{\chi}} - \overline{\sigma}^{-1} \partial_\theta \overline{\hat{Q} \sigma \hat{\chi}}\,.$$

Here, hats denote deviations from the mass-weighted zonal mean (e.g. $\hat{\chi} \equiv \chi - \overline{\chi}^*$). This is a slightly modified version of that given in Andrews et al. (1987), formulated here for the mass-weighted tracer mixing ratio. The effective vertical transport velocity in this case ($Q_{\mathrm{eff}}$) results from:

$$\partial_t \overline{\chi}^* + Q_{\mathrm{eff}} \partial_\theta \overline{\chi}^* = 0\,,$$

hence:

$$Q_{\mathrm{eff}} = \overline{Q}^* + (\partial_\theta \overline{\chi}^*)^{-1} \left( \overline{v}^* \partial_y \overline{\chi}^* + \overline{\sigma}^{-1} \partial_y \overline{\hat{v} \sigma \hat{\chi}} + \overline{\sigma}^{-1} \partial_\theta \overline{\hat{Q} \sigma \hat{\chi}} \right)\,.$$

In this case, assuming quasi-adiabatic mixing processes ($\hat{Q} \approx 0$, e.g. due to Rossby and gravity waves in the horizontal and vertical direction, respectively) and neglecting horizontal advection, horizontal mixing is the primary process that leads to differences between $\overline{Q}^*$ and $Q_{\mathrm{eff}}$.

Our estimates of $Q_{\text{eff}}$ from MLS agree roughly with diabatic heating rate estimates in the TTL and lower stratosphere (e.g. Fu et al., 2007; Wright and Fueglistaler, 2013), indicating that horizontal mixing does not play a big role in the observed tape recorder signal (cf. Ploeger et al., 2012). The difference between $\overline{Q}^*$ and $Q_{\text{eff}}$ is substantial in ERA-i, however, indicating excessive horizontal dispersion in the lowermost stratosphere.

Further insight into the role of vertical mixing may be obtained by comparing the effective vertical transport velocities in pressure and isentropic coordinates. Specifically, an approximate expression relating their difference to the vertical eddy tracer flux may be derived (outlined in the appendix):

$$\overline{\omega'\chi'} \approx \left[ \omega_{\text{eff}} - Q_{\text{eff}} \left(\partial_p \overline{\theta}\right)^{-1} \right] \frac{(\partial_{\bar{\theta}} \overline{\chi})^2}{\partial_{\bar{\theta}\bar{\theta}} \overline{\chi}} \, .$$

The factor outside the square brackets involves derivatives of the mean tracer mixing ratio with respect to the mean poten-
tial temperature, where both means are taken in pressure coordinates. This expression suggests that differences between the effective vertical transport velocities in the pressure versus isentropic coordinates are directly related to vertical mixing.

Figure 9 shows vertical profiles of this approximate vertical eddy flux of water vapor for DJF and JJA. The flux is predominantly negative in the lowermost stratosphere (in pressure coordinates), indicating the expected upward eddy transport from high to low background concentrations in height coordinates. This may serve as a sanity check that the above approxima-
tion gives physically reasonable results. The vertical gradient of the shown eddy flux ($\partial_p \overline{\omega'\chi'}$) confirms that vertical mixing contributes of the order of $10^{-3}$ to $10^{-2}$ ppmv/day to the overall water vapor tendency just above the tropical tropopause. However, the tendencies resulting from these eddy flux estimates only agree to within a factor of 10 with those derived from our 1-d model (cf. Fig. 7) and during DJF even have the opposite sign. This indicates that the approximations going into our eddy flux estimate at best provide qualitative results, although uncertainties also exist with our 1-d model results. Nevertheless,
given the lack of observational estimates of vertical eddy tracer fluxes on a zonal-mean scale, our approach, which at its heart takes advantage of comparing tracer evolutions in pressure and isentropic coordinates, may prove useful when applied to future higher resolution data sets.

## 5.2   One-dimensional transport modeling

In section 4.2 we found that a successful simulation of the water vapor tape recorder signal in pressure coordinates requires
strongly enhanced values for vertical mixing. A different story emerges when simulating the tape recorder signal in isentropic coordinates. In this case, the original transport parameters as obtained in Mote et al. (1998), translated into isentropic coordinates (cf. also Sparling et al., 1997), lead to a successful simulation matching the observations (with a score of $\sim 90\%$, shown in Figure 11). The corresponding time tendencies at 400 K (roughly corresponding to 80 hPa), shown in Figure 12, reveal that the total tendency is explained almost entirely by the contributions due to vertical advection ($\sim$ diabatic heating, red line) and
horizontal mixing (green), with the former dominating throughout NH winter and the latter dominating through NH spring and early summer.

Figure 10a shows the scores for a range of parameter combinations at 400 K for MLS, similar to Fig. 5. The range of high-scoring solutions is narrower than in pressure coordinates. Other than the reference/control set of parameters ($a = b = c = 1$)

we also find maximum scores for the case of no vertical mixing ($b = K = 0$) and control values for vertical advection and horizontal mixing ($a = c = 1$), and for the case of control value for vertical mixing ($b = 1$) and reduced vertical advection and horizontal mixing ($a = c = 0.5$). Overall, vertical mixing plays a smaller role in isentropic coordinates compared to pressure coordinates. This is expected based on the assumption that vertical mixing takes place quasi-adiabatically (see discussion in previous section).

Simulating the ERA-i tape recorder in isentropic coordinates requires increased strengths of the transport contributions (Fig. 10b). A factor of 2-3 increase in vertical advection and horizontal mixing compared to the control values together with an increase by at least a factor of 4 in vertical mixing leads to maximum scores ($> 90\%$). The increase in vertical advection points once more to biases in diabatic heating rates in ERA-i (presumable due to longwave cloud radiative biases in the TTL, as stated earlier). The increase in vertical mixing indicates excessive dispersion even in isentropic coordinates. We have found, however, that large changes in vertical mixing strength only lead to small changes in the simulated tape recorder signal (cf. that vertical gradients in the score distribution in Fig. 10b are much smaller than horizontal gradients), indicating that it is not very sensitive to this transport contribution.

## 6 Discussion

We have employed two methods to study transport contributions to the water vapor tape recorder signal in the tropical lowermost stratosphere: inferred effective vertical transport velocities and simple 1-d modeling in pressure and isentropic coordinates, respectively. Both methods indicate a significant role of vertical mixing in transport near the tropical tropopause. Our effective vertical transport velocity is larger than residual circulation upwelling, indicating additional vertical transport due to mixing. Our 1-d model setup is in principle identical to that used in Mote et al. (1998), with the important modification of seasonal dependency in the transport parameters. Residual circulation tropical upwelling is known to be much weaker during NH summer compared to NH winter (e.g. Rosenlof, 1995). Using annual mean vertical advection as in Mote et al. (1998) therefore artificially enhances its contribution to the total vertical transport during NH summer. It is in particular during NH summer then, where vertical mixing (parameterized as diffusion) plays a dominant role in the upward transport of water vapor, although we have found it to play a significant role throughout the year. Our most successful simulations of the observed tape recorder signal at 80 hPa using our modified idealized 1-d transport model incorporated a vertical diffusivity multiple times stronger compared to the control Mote et al. (1998) setting.

As a caveat to our results we stress that Aura MLS' vertical resolution of $\sim 3$ km, and even HALOE's doubled resolution, are coarse relative to the structures of interest in the lowermost tropical stratosphere. Our results are not qualitatively different between the two satellite products (and time periods). Higher resolution data sets are needed to conclude more definitively about the role of vertical mixing in tracer transport in this region. Nevertheless, it is instructive to note that vertical mixing alone can create a fairly realistic tape recorder signal using the 1-d model (not shown). To the extent that vertical mixing plays an important role in tropical lower stratospheric transport, the term "tape recorder", which refers more accurately to slow vertical advection, is misleading, at least near the tropopause (the same is true if horizontal mixing is important).

One potential drawback from our model setup is the neglect of the sink associated with explicit dehydration near the tape head (at the local cold point tropopause). Although even the lowest cold point pressures are generally higher than our lowest analyzed pressure level of 80 hPa (e.g. Seidel et al., 2001), the relatively large MLS averaging kernel of $\sim 3$ km means that e.g. the 100 hPa level still contributes 20% to the diagnosed 80 hPa level[3]. This means that some of the dehydration happening at

the local cold point tropopause will be projected onto the 80 hPa MLS level. In fact, Fig. 2 shows that the absolute minimum in MLS' lower stratospheric water vapor is diagnosed at 80 hPa in February, which suggests that part of the MLS signal during boreal winter at this level is due to dehydration. This may be less of an issue with the finer resolution HALOE data set (cf. Fig. S1 in supplement), although it also shows an absolute minimum during February at 80 hPa (but less pronounced compared to MLS).

To test whether dehydration can have a significant effect on our results, we have repeated our 1-d transport model calculation with a prescribed sink term ($S < 0$ in Eq. 1) (not shown). We used a seasonal functional form of a sine wave with strongest amplitude at 100 hPa that decays exponentially toward lower pressures and is set to zero at and above 70 hPa. We assumed that strongest dehydration of $S = -0.05$ ppmv/day happens at 100 hPa in January[4] and that $S = 0$ in July. This calculation with prescribed dehydration results in a more successful simulation of the water vapor evolution during boreal winter (as

expected – our simulation shown in Fig. 6 shows a moist bias during this season). The water vapor evolution during boreal summer, however, becomes less realistic: the dry bias already evident without dehydration (Fig. 6) generally increases, due to the propagation of the now dryer boreal winter signal into boreal summer. We therefore conclude that while the neglect of dehydration in our presented 1-d transport model results may explain the moist bias during boreal winter and may question the diagnosed strength of vertical mixing in that season, it does not improve the overall simulation of the water vapor evolution

throughout the year. In particular, dehydration tends to increase the dry bias during boreal summer, which would then demand an even greater contribution to the tape recorder signal due to mixing.

One possible reason for the dry bias during boreal summer is the neglect of the potential contribution by convective hydration (due to overshooting convection, e.g. Corti et al., 2008). Estimates of this contribution for the tropics-mean are difficult and so it is hard to say something more definitive about it. However, hydration due to convective overshooting essentially represents

vertical mixing (of total water, with subsequent evaporation of the condensate) and is therefore partially accounted for by the vertical mixing term in our simple transport simulations. Convection tends to reach deeper during boreal winter (e.g. Chae and Sherwood, 2010), which is consistent with our prescribed seasonality for $K$. Note that the vertical mixing tendency is a function of both $K$ and the vertical curvature of water vapor – it is the latter that peaks during boreal summer causing the strongest vertical mixing tendency during that season.

The influence by dehydration would be expected to vanish at levels above 80 hPa. We have also applied our 1-d transport model to these higher levels (not shown) and still find a significant impact by vertical mixing in pressure coordinates, although its amplitude decreases with height. For example, the top-scoring solution near 70 hPa uses $2 \times K_{ctrl}$ (i.e. half of that at 80 hPa)

---

[3]more precisely the output level is at 82.5 hPa

[4]This dehydration strength is consistent with that inferred from Lagrangian transport calculations a la Ploeger et al. (2012), Felix Ploeger, personal communication, 2016.

and the control settings for vertical advection and dilution. This supports our conclusion that vertical mixing is likely more important than previously estimated, although higher resolution data sets are needed to confirm this.

Support for the importance of vertical mixing in shaping the tape recorder signal also comes from comparing pressure and isentropic coordinates. We obtain physically reasonable differences between these coordinates. To the extent that vertical mixing involves primarily quasi-adiabatic processes (e.g. overshooting convection or breaking gravity waves) it is implicit in isentropic coordinates. It should therefore be less strong relative to other transport contributions when diagnosed in these co-ordinates and this is confirmed by our results based on both MLS and ERA-i. In fact, the observed tape recorder signal could be successfully simulated with our simple 1-d transport model using the control parameter settings translated into isentropic coordinates. Our results for these coordinates, including the importance of horizontal mixing for lowermost stratospheric transport, are also consistent with previous findings in the literature (e.g. Ploeger et al., 2012). The contribution from dehydration (or any other sources/sinks) would be expected to be largely independent of the coordinate system used, hence it would show up very similarly in both pressure and isentropic coordinates. The fact that we find vertical mixing to be much more important in pressure coordinates, but not so much in isentropic coordinates, then speaks against it being artificially enhanced due to the neglect of sources or sinks.

Another advantage of isentropic coordinates is that horizontal mixing, which is primarily due to Rossby wave breaking taking place along isentropes, is represented more dynamically consistently. It is conceivable that some of this mixing gets mapped into the vertical (due to undulating isentropic surfaces) when diagnosed in pressure coordinates. The simple 1-d formulation of our transport model (as in Mote et al., 1998) may misrepresent horizontal mixing, such that part of our diagnosed vertical mixing in fact represents masked horizontal mixing. Future work is required to shed more light on this caveat.

Data assimilation as used in reanalyses is known to cause spurious dispersion in the lower stratosphere (e.g. Schoeberl et al., 2003) and this most likely explains why our results indicate strongly enhanced vertical and horizontal mixing in ERA-i relative to observations. Effective vertical transport velocities inferred from the water vapor tape recorder signal are 3-4 times greater in ERA-i than in MLS or HALOE. These transport velocities are also significantly greater than ERA-i's residual circulation upwelling, suggesting that tropical lower stratospheric transport in ERA-i does not behave like a tape recorder. We find in particular the vertical mixing to be excessive in ERA-i, and this makes sense given the strong vertical gradient of water vapor near the tropical tropopause.

Another indicator for spurious transport caused by data assimilation in ERA-i is that the transport contributions inferred from the free-running climate model GEOS CCM are much more in alignment with the MLS or HALOE observations. We have also simulated GEOS' tape recorder signal using our idealized 1-d transport model and found similar transport parameter settings for the highest scoring simulations as in MLS and HALOE (not shown). Preliminary simulation results using other CCMs, however, show a range of vertical diffusivities suggesting that vertical mixing plays a more significant role in some models. Vertical diffusion likely also results numerically due to the limited resolution in the models, which might lead to numerical dissipation of waves as they propagate through the tropical tropopause.

Overall, our results confirm that transport in the tropical lowermost stratosphere is complicated with significant roles played by vertical advection, vertical mixing, and horizontal mixing. Vertical advection (= residual circulation upwelling) and hori-

zontal mixing are both to a large extent created by extratropical (Rossby) wave driving. Vertical mixing, on the other hand, is created by small scale processes, e.g. associated with overshooting convection or breaking gravity waves. The details of how these processes give rise to the inferred mixing are presently unclear. Kuang and Bretherton (2004) found vertical eddy heat fluxes due to overshooting convection to be of leading order importance in determining the temperature structure around the cold point tropopause in their small-domain cloud model study. Gravity wave breaking via convective instability has been shown to not be very effective in mixing heat and constituents vertically (e.g. Fritts and Dunkerton, 1985; Coy and Fritts, 1988). Inertio-gravity waves, on the other hand, tend to break down more preferentially due to shear instability (e.g. Dunkerton, 1984; Lelong and Dunkerton, 1998) and may give rise to substantially stronger mixing. Non-linear effects due to the strong stratification jump at the tropical tropopause and the associated tropopause inversion layer (Grise et al., 2010) may also give rise to enhanced vertical mixing in this region.

Due to the involved small horizontal and/or vertical scales, vertical mixing is much less well constrained in global models, but might contribute to variability and change from seasonal to centennial time scales. Given the importance of stratospheric water vapor for climate, it is important to better constrain the transport processes shaping the tape recorder signal near its base just above the tropical tropopause.

## Appendix A: Effective velocity comparison between pressure and isentropic coordinates

Neglecting the local time-tendency of zonal mean potential temperature, zonal mean diabatic heating is approximately given by:

$$\overline{Q} \approx \overline{\omega}^* \partial_p \overline{\theta} + \partial_p \overline{\omega'\theta'} \,,$$

where the last term may be thought of as representing the effects of vertical mixing. Assuming that the $\theta$-perturbations are primarily created by quasi-adiabatic vertical displacements (e.g. associated with gravity waves) acting on the background gradient, we can write:

$$\theta' \approx -\xi \partial_p \overline{\theta} \,,$$

where $\xi$ is the vertical displacement in pressure coordinates. Similarly, perturbations in a quasi-conserved tracer can be written:

$$\chi' \approx -\xi \partial_p \overline{\chi} \quad \Rightarrow \quad \theta' \approx \chi' \frac{\partial_p \overline{\theta}}{\partial_p \overline{\chi}} \,.$$

This allows us to write the vertical eddy heat flux as:

$$\overline{\omega'\theta'} \approx \overline{\omega'\chi'} \frac{\partial_p \overline{\theta}}{\partial_p \overline{\chi}} \quad \Rightarrow \quad \overline{Q} \approx \overline{\omega}^* \partial_p \overline{\theta} + \partial_p \left( \overline{\omega'\chi'} \frac{\partial_p \overline{\theta}}{\partial_p \overline{\chi}} \right) \,.$$

Now, assuming that the effective vertical transport velocity for $\chi$ is primarily composed of a residual circulation contribution and vertical mixing:

$$\omega_{\text{eff}} \approx \overline{\omega}^* + \partial_p \overline{\omega'\chi'} (\partial_p \overline{\chi})^{-1} \,,$$

we can insert $\overline{\omega}^*$ from the expression for $\overline{Q}$ to give:

$$
\begin{aligned}
\omega_{\text{eff}} \quad &\approx \quad \overline{Q} \, (\partial_p \overline{\theta})^{-1} - \partial_p \left( \overline{\omega'\chi'} \frac{\partial_p \overline{\theta}}{\partial_p \overline{\chi}} \right) (\partial_p \overline{\theta})^{-1} + \partial_p \overline{\omega'\chi'} (\partial_p \overline{\chi})^{-1} \\
&= \quad \overline{Q} \, (\partial_p \overline{\theta})^{-1} - \overline{\omega'\chi'} (\partial_p \overline{\theta})^{-1} \partial_p \left( \frac{\partial_p \overline{\theta}}{\partial_p \overline{\chi}} \right) \\
&= \quad \overline{Q} \, (\partial_p \overline{\theta})^{-1} + \overline{\omega'\chi'} \frac{\partial_{\bar{\theta}\bar{\theta}} \overline{\chi}}{(\partial_{\bar{\theta}} \overline{\chi})^2} \,,
\end{aligned}
$$

where the last step uses $\partial_p = \partial_p \overline{\theta} \, \partial_{\bar{\theta}}$. If $\overline{Q} \approx Q_{\text{eff}}$ (neglecting the horizontal transport contribution and still assuming quasi-adiabatic eddies) this provides an estimate of the vertical eddy flux of the tracer $\chi$:

$$\overline{\omega'\chi'} \approx \left[ \omega_{\text{eff}} - Q_{\text{eff}} \, (\partial_p \overline{\theta})^{-1} \right] \frac{(\partial_{\bar{\theta}} \overline{\chi})^2}{\partial_{\bar{\theta}\bar{\theta}} \overline{\chi}} \,.$$

*Acknowledgements.* This work has been supported by the US National Science Foundation's Climate Dynamics Program under grant #1151768. We acknowledge the criticism by one anonymous reviewer, which sparked the discussion of the potential role of dehydration in section 6. The comments by another anonymous reviewer helped to clarify many aspects of our manuscript. We further thank Timothy Dunkerton for constructive criticism and for sharing his insights into the transport processes shaping the tape recorder signal. Helpful comments on an earlier version were provided by Felix Ploeger.

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

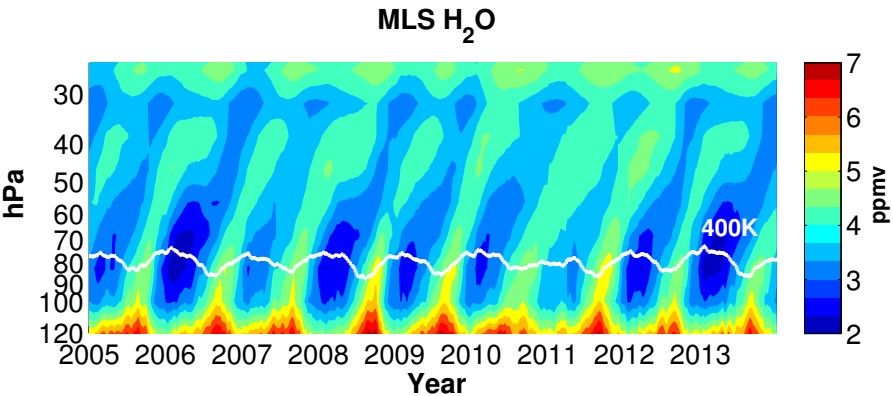

**Figure 1.** Zonal-mean tropical (10°S-10°N) tape recorder signal of water vapor (colored mixing ratio in ppmv) from MLS observations. The white line marks the 400 K isentrope for reference.

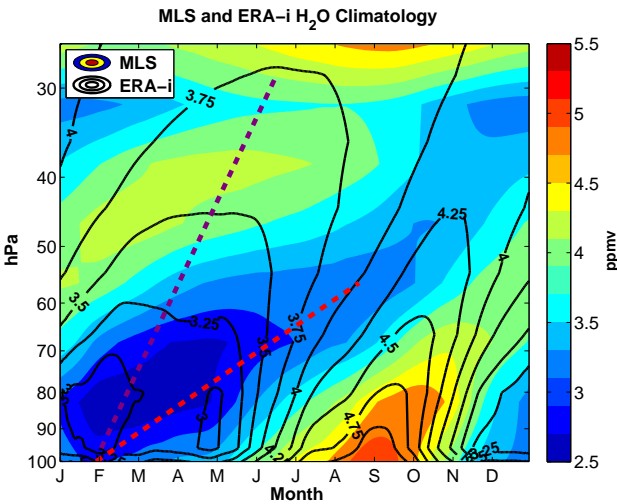

**Figure 2.** Climatological zonal-mean tropical (10°S-10°N) tape recorder signal (water vapor mixing ratio in ppmv) based on MLS (colors) and ERA-i reanalysis (black contours). The red and purple dotted lines roughly indicate the evolution of the dry minima with time for MLS and ERA-i, respectively.

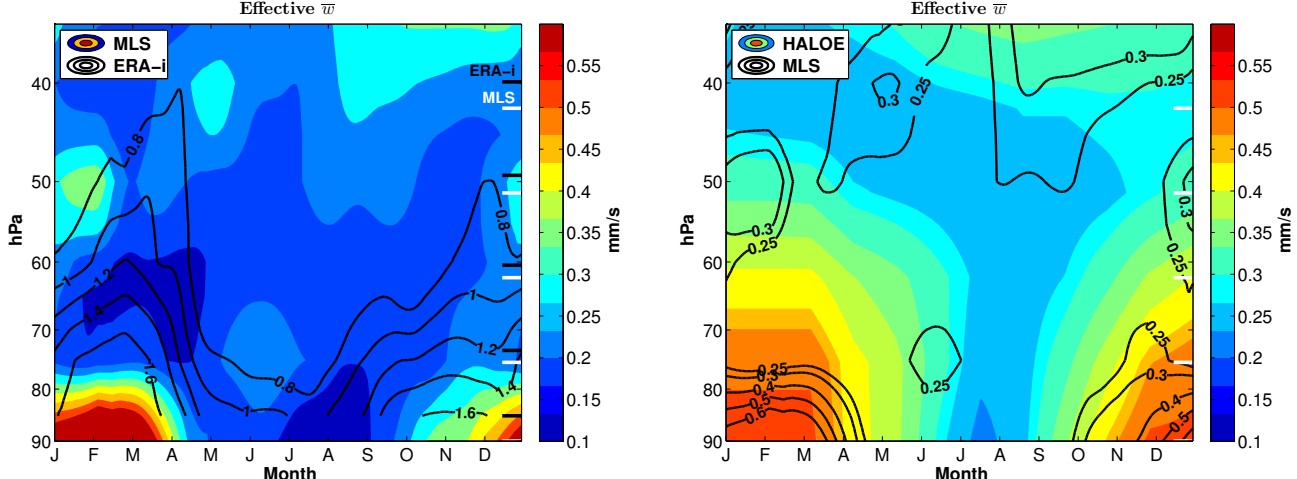

**Figure 3.** Effective vertical log-pressure transport velocities (mm/s, converted from pressure velocities by multiplying by $-H/p$, with $H = 7$ km) based on the phase-lagged correlation method (see text). Left: MLS observations (colors) vs. ERA-i reanalysis (black contours – note the different magnitude). Right: HALOE (colors) vs. MLS (black contours, same as colors on the left). Midpoint levels used for lag-correlations are indicated as white (MLS, HALOE) and black (ERA-i) bars on the right of each panel.

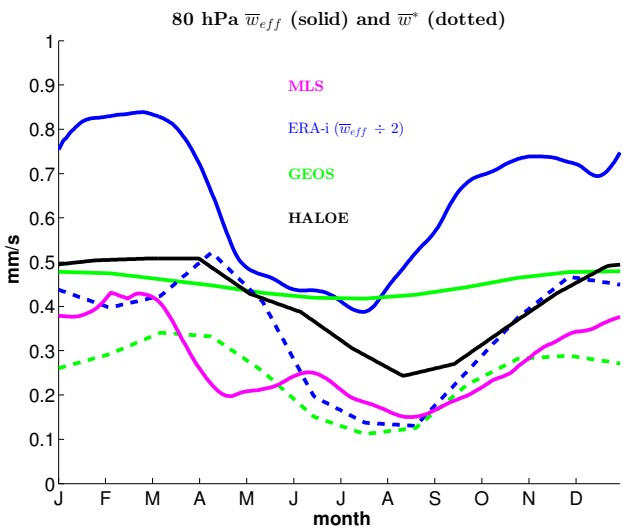

**Figure 4.** Effective vertical log-pressure transport velocities at 80 hPa (solid, converted from pressure velocities using $H = 7$ km) compared to TEM vertical residual velocities (dashed) from ERA-i and the GEOS-CCM. The ERA-i effective velocity has been divided by 2 to fit within the shown axis range.

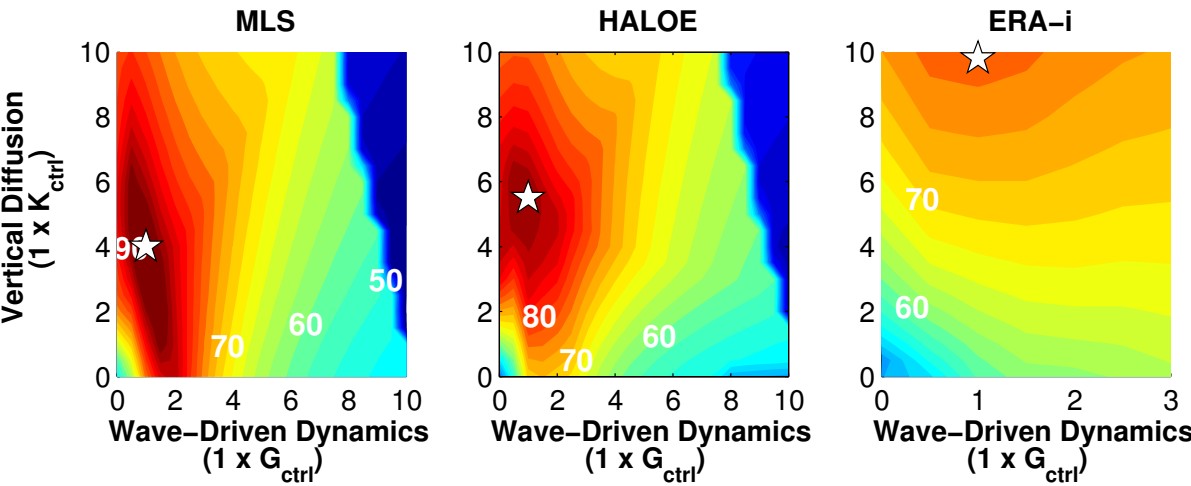

**Figure 5.** Percentage total scores (see text for details) of the synthetic MLS (left), HALOE (middle), and ERA-i (right) tape recorders at 80 hPa. White stars mark the combinations producing the highest scores.

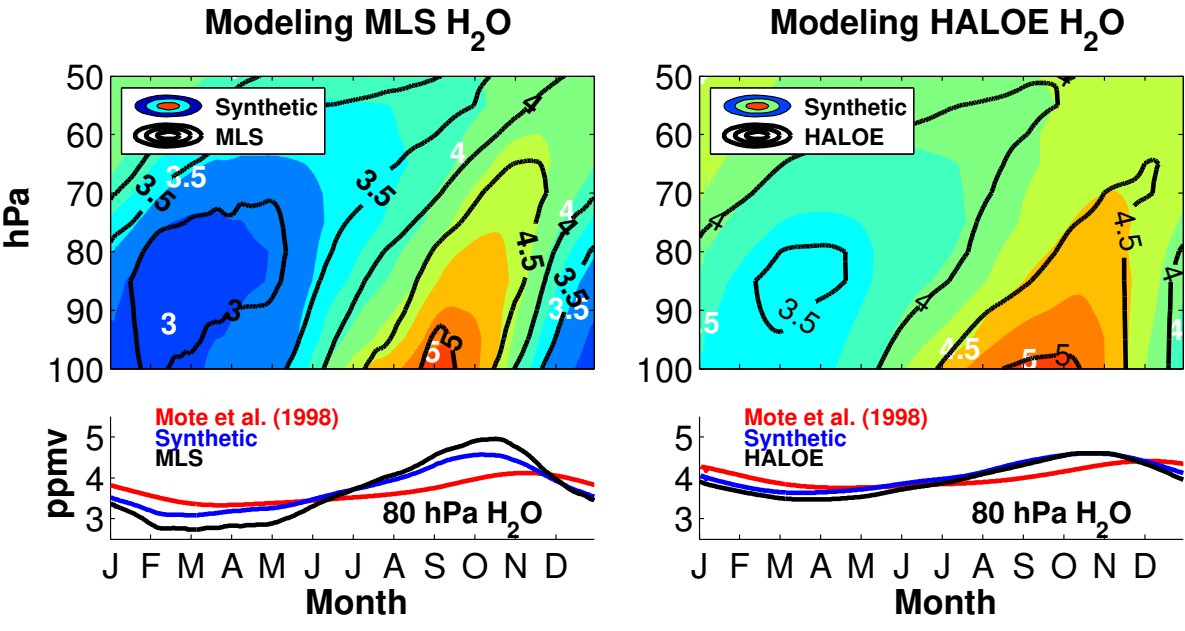

**Figure 6.** Best synthetic 1-d transport model solutions of the MLS (left, a=c=1, b=4) and HALOE (right, a=c=1, b=5.5) tape recorders in pressure coordinates (corresponding to the white stars in the left and middle panels of Fig. 5, respectively). The bottom panels show the annual cycle of the water vapor mixing ratio at 80 hPa produced using the Mote et al. (1998) control values (red), the above synthetic values (blue), and the MLS and HALOE observations (black).

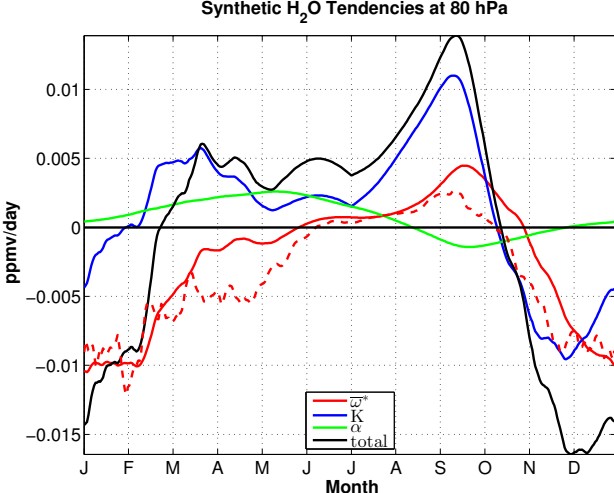

**Figure 7.** Contributions to the water vapor tendency (ppmv/day) at 80 hPa from the best synthetic 1-d transport model solution for MLS (a=c=1, b=4, corresponding to the white star in the left panel of Fig. 5). The red dashed line shows the tendency due to the vertical residual velocity from ERA-i.

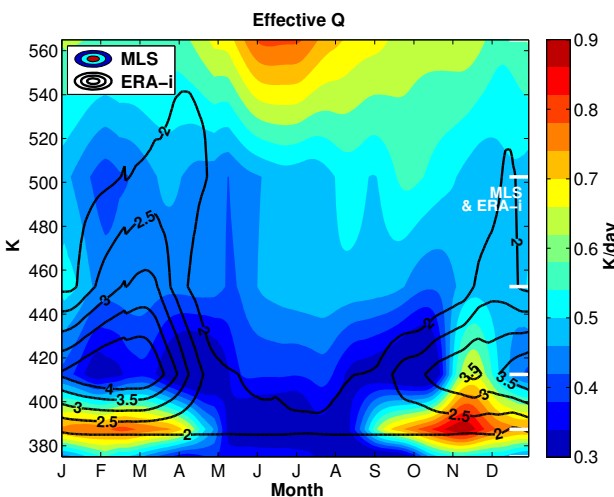

**Figure 8.** Effective vertical transport velocities in isentropic coordinates (effective diabatic heating rate, K day$^{-1}$) based on the phase-lagged correlation method. Colors: MLS observations; black contours: ERA-i reanalysis (note different magnitude). Midpoint levels used for lag-correlations are indicated as white bars on the right (same for MLS and ERA-i).

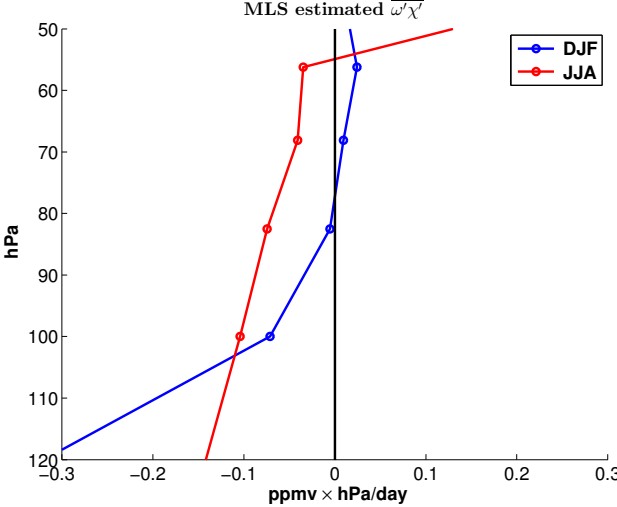

**Figure 9.** Estimated vertical eddy flux of water vapor based on the difference of MLS effective vertical transport velocities between pressure and isentropic coordinates (see text for details).

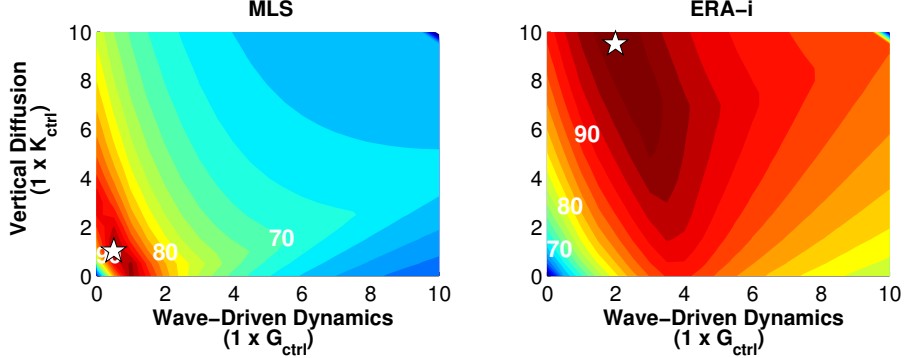

**Figure 10.** Total scores (%) of the synthetic MLS and ERA-i tape recorders at 400 K.

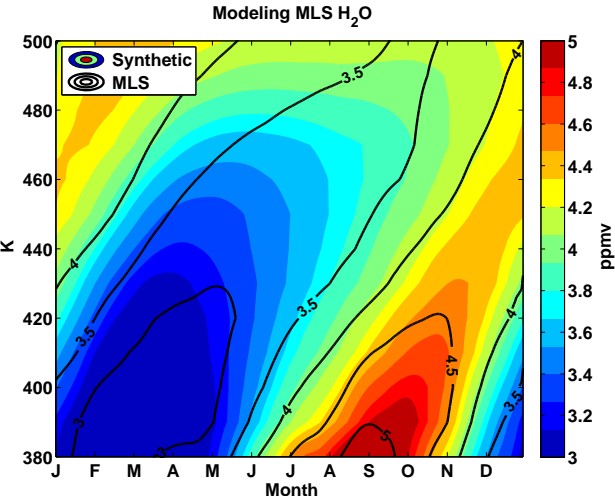

**Figure 11.** Best synthetic 1-d transport model solution (color shading, a=b=c=1, corresponding to white star in left panel of Fig. 10) of the MLS water vapor tape recorder signal (black contours for reference) in isentropic coordinates.

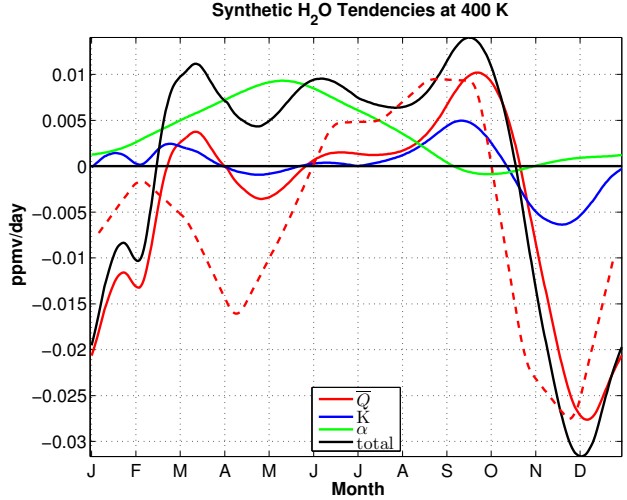

**Figure 12.** Contributions to the water vapor tendency (ppmv/day) at 400 K from the best synthetic 1-d transport model solution for MLS (a=b=c=1, corresponding to the white star in the left panel of Fig. 10). The red dashed line shows the tendency due to vertical advection (diabatic heating) from ERA-i.