# Peer review of "Role of vertical and horizontal mixing in the tape recorder signal near the tropical tropopause"

_Atmospheric Chemistry and Physics, 2016_

## Referee Comment (RC1) · Anonymous Referee #1 · 31 May 2016

This paper analyzes the seasonal cycle of tropical stratospheric water vapor based on satellite observations from MLS and an idealized tracer continuity equation (Eq. 1, with parameterized horizontal and vertical mixing terms). Comparisons are also made for water vapor in the ERAinterim reanalysis, which is completely a model result (no stratospheric water vapor data are assimilated in ERAinterim). The idealized model is used to fit the vertical propagation of the water vapor minimum derived from MLS data, assuming zero sources/sinks in Eq. 1 (i.e. changes above 100 hPa are interpreted as a combination of vertical advection and horizontal/vertical mixing above that level). Much of the focus is on variability at 80 hPa, and results are interpreted as demonstrating large amounts of vertical diffusion in the lower stratosphere (many times larger

than previous estimates). However, my opinion is that the authors have neglected to consider the important effects of explicit dehydration for the 80 hPa level (the S term in Eq. 1). The tropical cold point tropopause is near 90 hPa, and the water vapor estimated from the ~3 km broad layer MLS retrievals at 83 hPa certainly includes the influence of dehydration near the cold point. Note that an absolute minimum in water vapor is seen near 83 hPa in ~March in Fig. 2; note also that the red dashed line in Fig. 2 does not accurately trace the water vapor minima in altitude. Interpreting variations at 80 hPa as solely resulting from vertical advection and mixing from the 100 hPa level is incorrect, resulting in the large (and likely unrealistic) derived values of vertical diffusion. The calculations at 80 hPa are incorrect because of the neglect of explicit dehydration; the model may work better at altitudes above the dehydration level, but that is not the focus here. There are other aspects of the results that I am not comfortable with, as the details of the isentropic calculations and effective transport velocity in the second half of the paper get very complicated and difficult to understand. However, this doesn't matter, as the same fundamental problem exists as in the pressure coordinate formulation. Overall I am unconvinced by the analyses presented in the paper, and I do not suggest publication of this paper in anything close to present form.

---

## Author Comment (AC1) · 8 Jun 2016

Reviewer 1 raises an issue, which according to her/him is so critical that it essentially invalidates all major findings in our manuscript. This issue corresponds to our neglecting of the source/sink term in the water vapor budget (e.g. Eq. 1). Specifically, the reviewer argues that explicit dehydration at the cold point tropopause plays an important role for MLS' water vapor at ∼80 hPa, due to the relatively broad retrieval over a ∼3 km layer (which at this level extends from ∼100-65 hPa, i.e. near its lower boundary includes the cold point tropopause).

We'd like to urge the reviewer to consider the following points and perhaps re-evaluate our manuscript accordingly:

First of all, we agree that there is likely a contribution from explicit dehydration to the water budget at 80 hPa, which we neglect. That this wasn't discussed in the submitted manuscript is an oversight on our part. Note that due to the seasonal cycle of the cold point pressure (closer to 90 hPa during boreal winter, closer to 100 hPa during boreal summer), this effect would be expected to be more significant during boreal winter.

However, it is quite unlikely that the neglect of explicit dehydration invalidates our major findings:

1) Our results are consistent between MLS and HALOE (as stated on line 6, page 12), the latter having a better vertical resolution ($\sim$1.5 km) and hence presumably less impact from sources/sinks at the tropopause.

2) Dehydration would produce an additional negative tendency in our budget, especially during boreal winter when the cold point is located higher. However, this would in turn demand a larger positive tendency from the other terms to compensate. This would therefore if anything result in an even larger contribution due to mixing than we diagnose (cf. Fig. 7, possibly a combination of vertical and horizontal mixing) — the opposite of what the reviewer claims.

3) Furthermore, we find that vertical mixing is most important during boreal summer when the contribution from vertical advection is too small to keep the tape recorder going (cf. first paragraph of discussion section). But during boreal summer the cold point is lower making the expected contribution from explicit dehydration smaller and therefore contradicting the reviewer's claim.

4) Note also that the lower panel in Fig. 6 shows that a) our synthetic solution does a much better job than Mote et al. at capturing the observed evolution, b) we tend to overestimate the observed values during boreal winter (consistent with the neglect of explicit dehydration), c) we tend to underestimate the observed values during boreal summer (so dehydration would if anything make the situation worse in that season). One possible reason for our bias during boreal summer is that we neglect the potential

contribution of convective hydration (due to overshooting convection, e.g. Corti et al. 2008). Estimates of this contribution for the tropics-mean are difficult and so it's hard to say something more definitive about it. Dessler et al. (2016) recently found indirect evidence that this contribution might be significant for future stratospheric water vapor trends.

5) We'd also like to stress again (as in the paper, e.g. lines 13-21 on page 12) that we obtain physically reasonable differences between pressure and isentropic coordinates. Specifically, vertical mixing does not play an important role in isentropic coordinates and our results for these coordinates are consistent with previous findings in the literature (e.g. Ploeger et al. 2012). However, the contribution from dehydration (or any other sources/sinks) should be largely independent of the coordinate system used, hence it would show up very similarly in both coordinates. The fact that we find vertical mixing to be much more important in pressure coordinates, but not so much in isentropic coordinates, then speaks against it being artificially enhanced due to the neglect of sources or sinks.

6) It's possible that the simple 1-d formulation of our model (as in Mote et al. 1998) misrepresents horizontal mixing and that part of our diagnosed vertical mixing in fact represents masked horizontal mixing (cf. line 19-21 on page 12). Hopefully future work can shed more light on this caveat.

The reviewer also indicates potential issues related to our isentropic coordinate results, but unfortunately doesn't provide any actual argument regarding those results. We appreciate the comment that our presentation is hard to follow / complicated at places and would welcome any specific remarks and suggestions so that we can try to improve the paper at those places.

A couple of other remarks:

- Yes, ERA-i doesn't assimilated water in the stratosphere. However, given how strong of a function of the cold point temperature it is, and given that temperatures are assimilated, ERA-i's stratospheric water vapor should not be considered to be unconstrained. In fact, our Fig. 2 shows that apart from the tape recorder seasonality (i.e. transport strength), ERA-i and MLS agree quite well in the stratosphere (in terms of overall absolute values).

- Please note that nowhere in the paper do we claim that we've found the final answers to the transport problem near the tropical tropopause, nor do we claim that we have 100% proof that vertical mixing is as strong as indicated by our results (e.g. statement on line 7, page 12). Rather, we present evidence that points to a potentially greater importance of vertical mixing for transport just above the tropical tropopause than previously assumed.

Should the editor allow us to submit a revised manuscript, we will include a discussion of sources/sinks along the lines of the above in the paper.

Reference used:

Corti, T., et al. (2008), Unprecedented evidence for deep convection hydrating the tropical stratosphere, Geophys. Res. Lett., 35, L10810, doi:10.1029/2008GL033641.

Dessler, A. E., H. Ye, T. Wang, M. R. Schoeberl, L. D. Oman, A. R. Douglass, A. H. Butler, K. H. Rosenlof, S. M. Davis, and R. W. Portmann (2016), Transport of ice into the stratosphere and the humidification of the stratosphere over the 21st century, Geophys. Res. Lett., 43, 2323–2329, doi:10.1002/ 2016GL067991.

Ploeger, F., P. Konopka, R. Müller, S. Fueglistaler, T. Schmidt, J. C. Manners, J.-U. Grooß, G. Günther, P. M. Forster, and M. Riese (2012), Horizontal transport affecting trace gas seasonality in the Tropical Tropopause Layer (TTL), J. Geophys. Res., 117, D09303, doi:10.1029/2011JD017267.

---

## Referee Comment (RC2) · Anonymous Referee #2 · 9 Jun 2016

The paper addresses an important topic on transport processes in the tropical lower stratosphere. More accurate simulations of chemistry and climate require improved understanding of the dynamics of this region, and this paper could represent a key contribution towards this effort.

Overall the paper is very well written and the figures are clear. My only major concern surrounds the analysis methodology, as noted below. Otherwise, there are a few minor issues and clarifications noted.

Major:

One primary conclusion of this paper, that vertical mixing in the tropical lower strato-

sphere must be four times larger than the value estimated by Mote et al (1998), rests on the 1-d model simulations. While the model is fairly simple conceptually and is likely to be a useful tool for this kind of analysis, the devil is in the details which are not completely explained, and the impact of the certain assumptions embedded in the model are not fully explored.

1. Seasonality is introduced in the parameters omega, K, and alpha by prescribing reductions and enhancements of 50% over the course of a seasonal cycle. There is no discussion of why a 50% variation is a valid assumption. Is there observational evidence to support fixing this amplitude? How are results impacted if one chooses, say 30% or 70% amplitudes? Are these prescriptions sinusoidal seasonal variations? For the phases of seasonal cycles, the paper has some discussion that justifies the choices based on models or observations; however, "boreal winter" and "boreal summer" are given instead of dates or months, which would be preferable. For example, saying that horizontal mixing maximizes during boreal summer likely refers to the July-August period and not the NH summer solstice. Gettelman et al (2011) discuss the importance of horizontal mixing during July-August (e.g. the Asian monsoon anticyclone).

2. The model's score is based on comparisons with the amplitude, phase, and annual mean of the observed water vapor mixing ratio at 80 hPa (or 400 K), but it is not clear how these are derived from the MLS data. Is this from a simple FFT analysis? If so, Fig 4 indicates that the seasonal variations are not exactly sinusoidal, so how does this impact the analysis if a different functional form is used, one that better simulates the seasonality of the effective transport velocity? In this regard, is there any explanation for why the MLS velocity in Fig 4 has a double minimum, or is the spring dip just noise?

3. Use of a constant, 7-km scale height to convert from pressure velocity: This is not appropriate for a couple of reasons. First, temperatures near 70 hPa are about 200-210 K in the tropical lower stratosphere, so that the scale height is closer to 6 km. Second, there is a well-documented seasonal cycle in temperature that causes variations of 3-4% in the scale height, and this should be included in the calculation of

effective transport velocities, particularly in examining there seasonal behavior.

Minor:

1. Abstract, lines 8-9: This seems to state that the seasonal cycle of residual velocity derived from MLS has a larger amplitude than that in ERA-i, which conflicts with results shown in Figure 4.

2. Abstract, lines 20-21: "as opposed to" implies an either/or scenario, whereas I think this paper finds that a combination of slow upward transport *and* rapid vertical mixing play a role in shaping the tape recorder signal.

3. p 3, first paragraph: The latitude averaging for MLS data should be presented here, along with a discussion/justification of the choice of latitude bounds (appears to be 10S-10N from figure captions).

4. p. 4, lines 30-32: As correctly noted, the effect of methane oxidation is primarily an additive constant. This can be easily accommodated by looking at anomalies for the MLS data analysis, or by a simple parameterization of "S" in equation 1 for the 1-d model. Thus, this reason alone does not seem to be a valid motivation for restricting the analysis to altitudes less than 21 km ($\sim$40 hPa).

5. p. 5, line 32: The midlatitude reference mixing ratio should be allowed to vary seasonally for a correct model simulation. If that is the case, it should be clearly stated here.

6. p. 8, lines 18-20: "while its phase relies more on string enough vertical advection and on allowing for transport seasonality" is unclear. Is this saying something about simulating the phase of the tape recorder? If so, what is "strong enough" and for which transports (advection, vertical mixing, or horizontal mixing) are the seasonality important?

7. p. 10, lines 7-12: A lot of the notation needs to be clarified in the equations, e.g., what do the hat symbols represent?

8. p. 10, lines 27-32, and Figure 9: First, it is not obvious why we should care much about vertical profiles of derived vertical eddy fluxes. The "sanity check" rationale is a stretch, as the vertical gradient only gives consistency with the 1-d model to within a factor of 10, and upon closer inspection, the negative tendency shown in Fig 7 for the vertical eddy mixing in boreal winter should correspond to a negative slope in Fig 9 for DJF at 80 hPa, which is clearly not the case. Thus, it appears that there are very large errors in the calculated eddy fluxes (perhaps as expected when taking differences between two quantities with large inherent uncertainties). A more robust discussion of the uncertainties in these results is warranted, along with a more complete analysis (e.g. comparison with previous studies, or what has been used in the past in 1-d models) of calculated eddy fluxes.

---

## Referee Comment (RC3) · Anonymous Referee #1 · 21 Jun 2016

I appreciate the response of the authors to my previous review, and think it is appropriate to reply in turn to several of their comments.

1) **Our results are consistent between MLS and HALOE (as stated on line 6, page 12), the latter having a better vertical resolution (~1.5 km) and hence presumably less impact from sources/sinks at the tropopause.**

HALOE data shows an absolute minimum in water vapor during boreal winter at 83 hPa (e.g. Mote et al, 1998, Plate 1), similar to MLS, and I would argue that the results from both satellites are influenced by dehydration near the cold point. Note, however, that the Mote et al 1998 paper utilizes an EOF reconstruction of the HALOE data to perform their calculations of diffusion and dilution, and this reconstruction has water vapor extrema at the lowest level (100 hPa), and hence avoids dealing with the relative minimum at 83 hPa.

2) **Dehydration would produce an additional negative tendency in our budget, especially during boreal winter when the cold point is located higher. However, this would in turn demand a larger positive tendency from the other terms to compensate. This would therefore if anything result in an even larger contribution due to mixing than we diagnose (cf. Fig. 7, possibly a combination of vertical and horizontal mixing) ˘AˇT the opposite of what the reviewer claims.**

The large vertical diffusion calculated in this paper results in a strong negative H2O tendency at 83 hPa during November-January (shown in Fig. 7). I believe this tendency is compensating for the explicit dehydration that was neglected in the idealized model (which would occur exactly at this time).

3) **Furthermore, we find that vertical mixing is most important during boreal summer when the contribution from vertical advection is too small to keep the tape recorder going (cf. first paragraph of discussion section). But during boreal summer the cold point is lower making the expected contribution from explicit dehydration smaller and therefore contradicting the reviewer's claim.**

Figure 7 shows that vertical mixing is strong during August-October and November-January (with opposite signs). I don't understand the derived August-October maximum (and can't think of a reasonable physical mechanism for this timing), but I agree it is probably not tied to explicitly neglecting dehydration.

4) **Note also that the lower panel in Fig. 6 shows that a) our synthetic solution does a much better job than Mote et al. at capturing the observed evolution, b) we tend to overestimate the observed values during boreal winter (consistent with the neglect of explicit dehydration), c) we tend to underestimate the observed values during boreal summer (so dehydration would if anything make the situation worse in that season). One possible reason for our bias during boreal summer is that we neglect the potential contribution of convective hydration (due to overshooting convection, e.g. Corti et al. 2008). Estimates of this contribution for the tropics-mean are difficult and so it's hard to say**

**something more definitive about it. Dessler et al. (2016) recently found indirect evidence that this contribution might be significant for future stratospheric water vapor trends.**

As noted in the response to (1) above, the Mote et al 1998 analysis focused on an effectively vertically smoothed H2O data set, without the absolute minimum of water vapor at 83 hPa, so comparisons with the current results at this level are not straightforward. Tropical convection extends to higher altitudes in boreal winter compared to boreal summer (e.g. Chae and Sherwood, JAS, 2010), so there is little reason to expect a stronger signal above the tropopause during summer.

**5) We'd also like to stress again (as in the paper, e.g. lines 13-21 on page 12) that we obtain physically reasonable differences between pressure and isentropic coordinates. Specifically, vertical mixing does not play an important role in isentropic coordinates and our results for these coordinates are consistent with previous findings in the literature (e.g. Ploeger et al. 2012). However, the contribution from dehydration (or any other sources/sinks) should be largely independent of the coordinate system used, mixing to be much more important in pressure coordinates, but not so much in isentropic coordinates, then speaks against it being artificially enhanced due to the neglect of sources or sinks.**

This may be a valid argument. However, if the model is inappropriate and the results are questionable in pressure coordinates (the native coordinates of the MLS retrievals), I cannot be convinced they are reasonable by comparison to isentropic coordinate calculations (derived from vertical interpolations of the pressure level data).

**6) It's possible that the simple 1-d formulation of our model (as in Mote et al. 1998) misrepresents horizontal mixing and that part of our diagnosed vertical mixing in fact represents masked horizontal mixing (cf. line 19-21 on page 12). Hopefully future work can shed more light on this caveat.**

I agree it may be difficult to separate horizontal mixing from vertical diffusion using this idealized model. However, the neglect of explicit dehydration is a more important problem at 80 hPa. This idealized model applies to transport above the altitude of dehydration, i.e. tracking the minimum water vapor from the dehydration level to higher altitudes. In the MLS (or HALOE) data, the minimum water vapor occurs at the 83 hPa level, so it should be reasonable to apply the model above that level. However, applying this model to lower altitudes (and neglecting a physically important term) leads to the conclusion that vertical diffusion is a dominant process influencing the 83 hPa level, and I believe this conclusion is incorrect.

**- Please note that nowhere in the paper do we claim that we've found the final answers to the transport problem near the tropical tropopause, nor do we claim that we have 100% proof that vertical mixing is as strong as indicated by our results (e.g. statement on line 7, page 12). Rather, we present evidence that points to a potentially greater importance of vertical mixing for transport just above the tropical tropopause than previously assumed.**

Carl Sagan noted that 'extraordinary claims require extraordinary evidence'. My opinion remains that the important new result here (large transport due to vertical diffusion in the lower stratosphere) is not supported by the analysis. I appreciate that the authors have put significant effort into this work. I suggest either explicitly including dehydration in the calculations (probably difficult to do in an accurate manner) or simply focus on the region above 83 hPa, where the idealized model is more appropriate.

---

## Referee Comment (RC4) · Anonymous Referee #2 · 1 Jul 2016

The discussion between reviewer #1 and the authors has been interesting as well as instructive, and I would like to comment on some of the topics. First, the neglect of dehydration should be an acknowledged drawback but it does not invalidate the analysis to the point that the paper should be rejected. It cannot be the root cause for finding very large vertical diffusion except during 3 months of the year.

By definition, dehydration is always a negative tendency in the water vapor budget. Thus, the vertical mixing term will be overestimated only when the mixing tendency is negative in the model, which is during Nov-Dec-Jan. For a full eight months out of the year (March-October), the mixing tendency is positive; neglecting dehydration would lead to an underestimation of mixing, so the resulting large vertical mixing values for

[Figure]

March-October are not an artifact from this.

A realistic accounting of dehydration in this simple model is probably not feasible. Moving the analysis to higher altitudes might help reduce any impact from dehydration, but there is also a need for understanding what is going on at 80 hPa and I would argue that the results for March-October are interesting enough in their own right. There is no need for "extraordinary claims" nor extraordinary evidence, but there is sufficient evidence for the potential importance of vertical mixing in the tropical lower stratosphere.

---

## Author Comment (AC2) · 3 Aug 2016

Reviewer #1 comments in plain font.
**Author response on 1st round in public discussion in bold font, new responses in blue italic font.**

Revised manuscript with highlighted changes attached as supplement.

*General comments:*

*The primary issue this reviewer sees with our results is the neglect of explicit dehydration in the 1-d transport model. In our initial reply to the reviewer (see public discussion) we present arguments (highlighted in bold font below) why we think that it's quite unlikely that the neglect of dehydration significantly influences our results.*

*As discussed in section 6 of the revised manuscript (3rd and 4th paragraph in that section), we have also performed sensitivity experiments with our 1-d transport model by incorporating prescribed amounts of dehydration at its lowest levels (see plots shown below). These experiments confirm our expectation that dehydration brings the simulated water vapor signal closer to the observed one during boreal winter. However, it actually degrades the simulation during boreal summer, even though we don't apply dehydration during that season. This is because the now drier signal during DJF is propagated somewhat into JJA. This would then require an even larger amount of mixing during JJA. Also, the overall agreement of the entire seasonal water vapor evolution is not much different from the simulation without any dehydration. We conclude that it's quite unlikely that the neglect of dehydration explains the diagnosed levels of mixing strength in our simulations.*

[Figure]

*We have furthermore followed the suggestion by the reviewer to analyze higher levels (above 80 hPa) where the effect of dehydration can be neglected (see added text in discussion section of revised manuscript – paragraph 5 in section 6). While the diagnosed strength of vertical mixing (diffusion) does decrease with altitude (as expected*

*physically – as one moves away from the convective tops and the tropopause with their associated turbulence and small scale wave activity), it is still significantly enhanced relative to the control value. At 70 hPa the top-scoring solution still uses 2 times the control value for K. In isentropic coordinates, however, the control values remain adequate, which again is consistent with our arguments related to the difference in coordinates used.*

*In summary, we appreciate the issue brought forward by the reviewer but feel that we have sufficient evidence to show that this issue is not severe enough to invalidate our main conclusions. We have incorporated additional discussion in section 6 of the revised manuscript, which we hope makes this section as a whole a more balanced discussion of the strengths and weaknesses of our approaches and conclusions.*

**Specific Comments**

**1) Our results are consistent between MLS and HALOE (as stated on line 6, page 12), the latter having a better vertical resolution (~1.5 km) and hence presumably less impact from sources/sinks at the tropopause.**

HALOE data shows an absolute minimum in water vapor during boreal winter at 83 hPa (e.g. Mote et al, 1998, Plate 1), similar to MLS, and I would argue that the results from both satellites are influenced by dehydration near the cold point. Note, however, that the Mote et al 1998 paper utilizes an EOF reconstruction of the HALOE data to perform their calculations of diffusion and dilution, and this reconstruction has water vapor extrema at the lowest level (100 hPa), and hence avoids dealing with the relative minimum at 83 hPa.

*While we agree that results from both satellites are influenced by dehydration, we expect the degree of that influence to be smaller for HALOE, due to its finer vertical sampling. As elaborated in our general comments above, we also agree that the minimum water vapor at 80 hPa during Feb-March is likely a signature of this dehydration influence (and we have added a remark in the discussion section of the revised manuscript). However, our sensitivity experiment shown in the general comments suggests that the overall influence by dehydration is small (although it does improve the simulation during the season where one would expect it – DJF).*

**2) Dehydration would produce an additional negative tendency in our budget, especially during boreal winter when the cold point is located higher. However, this would inturn demand a larger positive tendency from the other terms to compensate. Thiswould therefore if anything result in an even larger contribution due to mixing than we diagnose (cf. Fig. 7, possibly a combination of vertical and**

**horizontal mixing) âˇAˇT the opposite of what the reviewer claims.**

The large vertical diffusion calculated in this paper results in a strong negative H2O tendency at 83 hPa during November-January (shown in Fig. 7). I believe this tendency is compensating for the explicit dehydration that was neglected in the idealized model (which would occur exactly at this time).

*See above. It's possible that our vertical mixing tendency is off during NDJ, due to the neglect of dehydration but that still only accounts for 25% of the year. Our sensitivity tests shown in the general comment suggests that the incorporation of dehydration doesn't change the diagnosed mixing strength much.*

**3) Furthermore, we find that vertical mixing is most important during boreal summer when the contribution from vertical advection is too small to keep the tape recorder going (cf. first paragraph of discussion section). But during boreal summer the cold point is lower making the expected contribution from explicit dehydration smaller and therefore contradicting the reviewer's claim.**

Figure 7 shows that vertical mixing is strong during August-October and November-January (with opposite signs). I don't understand the derived August-October maximum (and can't think of a reasonable physical mechanism for this timing), but I agree it is probably not tied to explicitly neglecting dehydration.

*We agree.*

**4) Note also that the lower panel in Fig. 6 shows that a) our synthetic solution doesa much better job than Mote et al. at capturing the observed evolution, b) we tend to overestimate the observed values during boreal winter (consistent with the neglect of explicit dehydration), c) we tend to underestimate the observed values during boreal summer (so dehydration would if anything make the situation worse in that season). One possible reason for our bias during boreal summer is that we neglect the potential contribution of convective hydration (due to overshooting convection, e.g. Corti et al. 2008). Estimates of this contribution for the tropics-mean are difficult and so it's hard to say something more definitive about it. Dessler et al. (2016) recently found indirect evidence that this contribution might be significant for future stratospheric water vapor trends.**

As noted in the response to (1) above, the Mote et al 1998 analysis focused on an effectively vertically smoothed H2O data set, without the absolute minimum of water vapor at 83 hPa, so comparisons with the current results at this level are not straightforward. Tropical convection extends to higher altitudes in boreal winter compared to boreal summer (e.g. Chae and Sherwood, JAS, 2010), so there is little reason to expect a stronger signal above the tropopause during summer.

*We agree and have modified the discussion of the potential role of convective hydration during summer in the revised manuscript.*

**5) We'd also like to stress again (as in the paper, e.g. lines 13-21 on page 12) that we**

**obtain physically reasonable differences between pressure and isentropic coordinates. Specifically, vertical mixing does not play an important role in isentropic coordinatesand our results for these coordinates are consistent with previous findings in the literature (e.g. Ploeger et al. 2012). However, the contribution from dehydration (or any other sources/sinks) should be largely independent of the coordinate system used,mixing to be much more important in pressure coordinates, but not so much in isentropic coordinates, then speaks against it being artificially enhanced due to the neglectof sources or sinks.**

This may be a valid argument. However, if the model is inappropriate and the results are questionable in pressure coordinates (the native coordinates of the MLS retrievals), I cannot be convinced they are reasonable by comparison to isentropic coordinate calculations (derived from vertical interpolations of the pressure level data).

*We don't understand this argument. The isentropic coordinates are derived from a consistent observational product (as opposed to e.g. incorporating temperatures from a reanalysis). The interpolation calculation is simple and straightforward. We maintain that we are able to reproduce findings from the past literature, which are physically reasonable, in isentropic coordinates, and that this supports the validity of our approach.*

**6) It's possible that the simple 1-d formulation of our model (as in Mote et al. 1998) misrepresents horizontal mixing and that part of our diagnosed vertical mixing in fact represents masked horizontal mixing (cf. line 19-21 on page 12). Hopefully future work can shed more light on this caveat.**

I agree it may be difficult to separate horizontal mixing from vertical diffusion using this idealized model. However, the neglect of explicit dehydration is a more important problem at 80 hPa. This idealized model applies to transport above the altitude of dehydration, i.e. tracking the minimum water vapor from the dehydration level to higher altitudes. In the MLS (or HALOE) data, the minimum water vapor occurs at the 83 hPa level, so it should be reasonable to apply the model above that level. However, applying this model to lower altitudes (and neglecting a physically important term) leads to the conclusion that vertical diffusion is a dominant process influencing the 83 hPa level, and I believe this conclusion is incorrect.

*See our general comments regarding results at higher levels and explicit incorporation of dehydration.*

---

## Author Comment (AC3) · 3 Aug 2016

Reviewer #2 comments in plain font.
**Author response in bold font.**
* * *
**We thank the reviewer for carefully reading our manuscript and pointing out a number of points that needed clarification. We specifically would like to thank this reviewer for sharing her/his perspective on the issue raised by reviewer 1.**

**We have incorporated more detail about the specifics of the model settings and its evaluation, along the lines of the reviewers' major comments. Our specific changes and responses to the reviewer are summarized below.**

**We also attached a revised manuscript with highlighted changes as supplement.**

Major:

One primary conclusion of this paper, that vertical mixing in the tropical lower stratosphere must be four times larger than the value estimated by Mote et al (1998), rests on the 1-d model simulations. While the model is fairly simple conceptually and is likely to be a useful tool for this kind of analysis, the devil is in the details which are not completely explained, and the impact of the certain assumptions embedded in the model are not fully explored.

1. Seasonality is introduced in the parameters omega, K, and alpha by prescribing reductions and enhancements of 50% over the course of a seasonal cycle. There is no discussion of why a 50% variation is a valid assumption. Is there observational evidence to support fixing this amplitude? How are results impacted if one chooses, say 30% or 70% amplitudes? Are these prescriptions sinusoidal seasonal variations? For the phases of seasonal cycles, the paper has some discussion that justifies the choices based on models or observations; however, "boreal winter" and "boreal summer" are given instead of dates or months, which would be preferable. For example, saying that horizontal mixing maximizes during boreal summer likely refers to the July-August period and not the NH summer solstice. Gettelman et al (2011) discuss the importance of horizontal mixing during July-August (e.g. the Asian monsoon anticyclone).

> **The following was motivation for choosing 50% seasonal variance in the 3 terms:**
>
> **For vertical advection we oriented ourselves at Rosenlof (1995) and Abalos et al. (2013). We feel that these and other references on the subject constrain the variations for vertical advection (w\*) strongly enough that 50% seems a solid choice. Also note, our results are very similar in terms of the w\* tendency in ERA-i (dashed red lines in Figure 7).**

For horizontal mixing ($\alpha$) we oriented ourselves at Gettelman et al. (2011) and Ploeger et al. (2012, see reference in revised manuscript). It is clear from these and other references in the literature that horizontal mixing is stronger during (late) boreal summer.

For vertical mixing (K) we primarily referred to Flannaghan and Fueglistaler (2014), who indicate more vertical mixing during DJF but the seasonal cycle amplitude is uncertain.

The 50% choice is admittedly less obvious for both mixing terms. We used it for simplicity but also note the following.

We tested the model without seasonality in the two mixing terms (but still with 50% seasonality in vertical advection) and the resulting scores for the MLS tape recorder (at 80 hPa) can be seen in the figure below. Removing the seasonal cycle in K and $\alpha$ results in needing 50% stronger w* for the top-scoring solution (which seems unrealistic based on literature listed above). The best simulations (>90%) still require amplifying K by at least a factor of two. There is a slight "fork" with the warm colors seen in the figure below, both requiring amplified K.

[Figure]

Overall, this and other tests we performed reveal that the seasonal cycle in vertical advection is most crucial and this is the one that is also best constrained by past literature. Our main qualitative result that vertical mixing is as important as vertical advection does not seem to be very sensitive to the choice in seasonality-strength in K and/or alpha. Incorporating a 50% seasonal cycle to the mixing terms slightly narrows down the solutions, perhaps bringing them closer to reality.

**Also, the cycles are based on a sine wave, and the peaks occur in the middle days of January and July – this has been clarified in the text.**

2. The model's score is based on comparisons with the amplitude, phase, and annual mean of the observed water vapor mixing ratio at 80 hPa (or 400 K), but it is not clear how these are derived from the MLS data. Is this from a simple FFT analysis? If so, Fig 4 indicates that the seasonal variations are not exactly sinusoidal, so how does this impact the analysis if a different functional form is used, one that better simulates the seasonality of the effective transport velocity? In this regard, is there any explanation for why the MLS velocity in Fig 4 has a double minimum, or is the spring dip just noise?

**Thanks for pointing out need for clarification. Correct, the phase is calculated using a simple FFT analysis. But the amplitudes are obtained from the minimum and maximum values. Clarifying sentence has been added.**

**We feel that the climatological seasonal evolution of water vapor is sufficiently sinusoidal (e.g. Fig. 6 bottom) that this simple FFT analysis to obtain the phase is adequate. Note that Fig. 4 is a plot of vertical velocities, not the water vapor evolution (the latter is used to obtain the score).**

**We believe that the spring dip in MLS effective vertical velocity in Fig. 4 is due to noise. By testing the wEff method on synthetic tape recorders with different vertical resolutions, we found that coarser resolutions resulted in more noise, especially for the transition between the wet and dry signals. Note added in section 4.1.**

3. Use of a constant, 7-km scale height to convert from pressure velocity: This is not appropriate for a couple of reasons. First, temperatures near 70 hPa are about 200-210 K in the tropical lower stratosphere, so that the scale height is closer to 6 km. Second, there is a well-documented seasonal cycle in temperature that causes variations of 3-4% in the scale height, and this should be included in the calculation of effective transport velocities, particularly in examining there seasonal behavior.

**We prefer to work with log-p coordinates, as this makes comparisons to models most straightforward (which usually run in p-coo.). This means that H needs to be a constant (no seasonal variations, otherwise we would not be working in a p-coo. anymore). We use H = 7 km simply because this seems to be the standard value that people use in the literature (and in text books, e.g. Andrews 1987), despite the fact (well-taken by reviewer) that 7 km is off in the tropical LS. We've included a clarifying comment in the manuscript and modified the Fig. captions.**

Minor:

1. Abstract, lines 8-9: This seems to state that the seasonal cycle of residual velocity derived from MLS has a larger amplitude than that in ERA-i, which conflicts with results shown in Figure 4.

**Thanks for pointing out - the sentence has been reworded.**

2. Abstract, lines 20-21: "as opposed to" implies an either/or scenario, whereas I think this paper finds that a combination of slow upward transport *and* rapid vertical mixing play a role in shaping the tape recorder signal.

**Our "as opposed to" refers to the term "tape recorder", for which we do in fact mean to imply an either/or scenario: if transport is dominated by slow (vertical) advection then "tape recorder" is a justifiable term, but if mixing plays an important role (regardless of how important advection still is) then the term "tape recorder" becomes misleading. So we wish to leave the sentence as is.**

3. p 3, first paragraph: The latitude averaging for MLS data should be presented here, along with a discussion/justification of the choice of latitude bounds (appears to be 10S-10N from figure captions).

**Thanks for pointing out lack of clarity. 10S-10N is a common choice for the inner tropics – in our case it makes sure we have sufficient sampling and cover the latitudinal variations in the location of maximum upwelling. We didn't find much sensitivity to making the latitude band slightly bigger (15S-15N). Text has been added in section 2 to clarify.**

4. p. 4, lines 30-32: As correctly noted, the effect of methane oxidation is primarily an additive constant. This can be easily accommodated by looking at anomalies for the MLS data analysis, or by a simple parameterization of "S" in equation 1 for the 1-d model. Thus, this reason alone does not seem to be a valid motivation for restricting the analysis to altitudes less than 21 km (~40 hPa).

**We agree with the reviewer and appreciate the idea how to circumvent the complications due to methane oxidation at higher levels. However, we are particularly interested in the region just above the tropopause, which has been less studied from a tape recorder perspective and where vertical mixing may play a bigger role. We agree that the way we stated our motivation is misleading and have reworded the statement accordingly.**

5. p. 5, line 32: The midlatitude reference mixing ratio should be allowed to vary seasonally for a correct model simulation. If that is the case, it should be clearly stated here.

**We have added to the text: it does vary seasonally.**

6. p. 8, lines 18-20: "while its phase relies more on string enough vertical advection and on allowing for transport seasonality" is unclear. Is this saying something about simulating the phase of the tape recorder? If so, what is "strong enough" and for which transports (advection, vertical mixing, or horizontal mixing) are the seasonality important?

**Thanks for bringing up need for clarification. Yes the statement refers to simulating the phase on its own. "Strong enough" refers to scores over 90% for each individual measures (phase, amplitude, and annual mean). We found that different swaths (of factors beyond the control) can satisfy those measures when assessing their scores individually. For example, the amplitude alone scored best with 3xK_ctrl while the phase alone scored best with a variety of factors (2-6xK_ctrl). However, the phase had a narrower swath of best simulations when analyzing it in terms of vertical advection (w*). It's also the seasonality of advection that matters most. Sentence has been reworded to clarify.**

7. p. 10, lines 7-12: A lot of the notation needs to be clarified in the equations, e.g., what do the hat symbols represent?

**Thanks for pointing this out; notation has been clarified in the revised text. However, beyond the hat symbols (and the primes earlier in the text), we didn't find any other notation that needed clarification. Overbars and asterisks had already been introduced after Eq. (2).**

8. p. 10, lines 27-32, and Figure 9: First, it is not obvious why we should care much about vertical profiles of derived vertical eddy fluxes. The "sanity check" rationale is a stretch, as the vertical gradient only gives consistency with the 1-d model to within a factor of 10, and upon closer inspection, the negative tendency shown in Fig 7 for the vertical eddy mixing in boreal winter should correspond to a negative slope in Fig 9 for DJF at 80 hPa, which is clearly not the case. Thus, it appears that there are very large errors in the calculated eddy fluxes (perhaps as expected when taking differences between two quantities with large inherent uncertainties). A more robust discussion of the uncertainties in these results is warranted, along with a more complete analysis (e.g. comparison with previous studies, or what has been used in the past in 1-d models) of calculated eddy fluxes.

**Fair enough, we agree with the reservation by the reviewer about this section. Our primary motivation to include it is that observational estimates of vertical eddy tracer fluxes on a zonal-mean scale are essentially non-existent. But they are required to be able to quantify more accurately the role of vertical mixing. Despite the large errors in our estimated fluxes, we feel it's useful to include these results as they might inspire future research in that direction. We are not aware that our theoretical approximate formula derived in the appendix has been pointed out or used before, so the hope is**

that it could be useful for future studies. At the least we feel that the idea to parse out information about vertical mixing by comparing pressure (or height) to isentropic coordinates is novel and the related theoretical discussion may be insightful to some readers.

The section has been revised, emphasizing the uncertainties more.

---

## Referee Report (RR1)

Second review of Glanville and Birner, "Role of vertical and horizontal mixing...",
revised manuscript

The revised manuscript addresses most of the concerns raised by this reviewer, except for
sensitivities from assuming 50% enhancements and reductions in the transport
parameters. Results presented in the authors' response give an indication of the
sensitivity to the seasonality in the two mixing terms, and this should be discussed in the
paper. But there is no corresponding result for the vertical advection.

While the assumption of a 50% variation is reasonable, it is hardly a certain value for the
seasonal amplitude of the vertical advection. Two references provided in support of this
50% value are not particularly definitive. Rosenlof (1995) found seasonality in the 70-
hPa mass flux, but there was a large variation in amplitude depending on year and
method, ranging from +-25% to +-35% by eye in Fig 11 of that paper. The w*
seasonality referenced from Abalos et al. (2013) is derived from a model and should not
be viewed as observationally constrained. On the other hand, Abalos et al. (2015)
compared w* from three different reanalyses. While the phases of seasonal variations
were in good agreement, they found a relatively large range in both the overall magnitude
of upwelling and in the seasonal amplitudes (cf. their fig 7).

The authors also point to the good agreement with the w* tendency from ERA-i (Figure 7
of the revised manuscript) as supporting their 50% value. While it is reassuring, this
agreement does not provide definitive support for a 50% seasonal amplitude; it has been
established (and noted in this paper) that the effective transport in ERA-i is twice as fast
compared to observations. If the magnitudes differ by this much, it is hard to have
confidence in precise seasonal amplitudes from reanalyses. I would not recommend
turning off seasonality completely, but as suggested previously, testing a range between
30% and 70% would not be unreasonable and should be explored.

---

## Author Response (AR2)

**Response to Reviewer Comments, revised version**

Based on the additional reviewer comments, in particular those of reviewer 3, we have implemented several modifications to our manuscript (see detailed comments below). Reviewer 3 strongly encouraged to include HALOE results and this advice has been followed: Figures 3-6 now show HALOE results alongside MLS results, while versions of Figures 2 and 7 showing HALOE results are provided as supplement.

**Reviewer 1:**

Some comments regarding my reasons for rejecting this paper:
My previous review and comments on this paper have focused on the lack of including dehydration in the idealized 1D model calculations, which may lead to spurious derived vertical diffusion near the cold point tropopause. The authors have addressed this criticism by including calculations with idealized dehydration during winter, but the strongest dehydration is specified at 100 hpa, not near the cold point (the minimum in observed water vapor during boreal winter is near 83 hPa in both MLS and HALOE retrievals). In the end the authors discount the importance of explicit dehydration from these tests because it "...does not improve the overall simulation throughout the year...and increases the dry bias during the summer". However, the summer results (at 80 hPa) seem unphysical to begin with (in my opinion – see below), so this is a poor argument for not including dehydration (which is the fundamental cause of the tape recorder to begin with).

To the extent that dehydration (or any other source / sink) is prescribed, i.e. does not depend on the simulated water vapor mixing ratio, the details of how it is prescribed should not matter much qualitatively – they will result in simple (possibly seasonally depend) offsets of the simulated tape recorder signal. In the extreme case one could simulate the entire tape recorder purely by specifically designed sources and sinks, without any transport contributions – in strong conflict with what is well-known about lower stratospheric transport. In our view, it is preferable then to not include any prescribed source/sink term (as we have done, following Mote et al.) and rather diagnose missing tendencies after the fact. We agree that the low bias of our simulated water vapor during boreal winter and spring is consistent with a missing sink (e.g. due to dehydration), as discussed in the manuscript.

Note that the bulk of dehydration happens below 80 hPa, and therefore is included in the lower boundary condition of our simulations. So, the statement that we didn't include any dehydration is false. After all, even the Mote et al. control setting (small vertical mixing) produces a qualitatively realistic looking tape recorder signal at 80 hPa. As an aside, dehydration is only one fundamental cause of the tape recorder, the other being vertical transport (~the movement of the tape).

While intuitively I don't like the neglect of an important term in the idealized model, it is the detailed output of the model that is problematic in my opinion. Fig. 7 shows vertical mixing maximizing with identical timing to vertical advection, with extrema during Nov-Dec and Aug-Oct (which they term 'summer'). This timing seems unphysical to me; why should vertical mixing be tied to mean vertical advection? And what is the physical mechanism for these seasonal maxima in derived mixing?

First of all, we disagree that the timing between the vertical advection and vertical mixing terms in Fig. 7 is "identical". For example, the strongest negative tendency due to vertical mixing happens in late November, 1-2 months before that of the vertical advection. Vertical mixing goes through zero tendency in early February when vertical advection is at its strongest. Second, even if both terms did have identical timing we don't see how that alone indicates anything "unphysical": 1) the mathematical relation between the two (based on the vertical gradient and vertical curvature, respectively) certainly allows for a physical relation (e.g. think of an exponential); 2) a vast range of atmospheric processes that have absolutely

no physical relation to each other still share similar seasonality (e.g. arctic sea ice extent and Colorado snow cover) – in general, common seasonality is a very poor indicator of direct physical relationships between two variables.

That said, we agree that our maximized tendency due to vertical mixing during August-September seems surprising at first. However, note that this doesn't indicate that more vertical mixing occurs during that time, but that it has a bigger effect on water vapor. The vertical mixing tendency is a function of the background vertical curvature of the water vapor profile. This curvature maximizes during August-September giving rise to a large vertical mixing tendency, even if the diffusivity is held constant throughout the year. In principle, the mixing tendency could be larger even if the amount of mixing is smaller (i.e. in our case smaller K during August-September), as long as the curvature contribution overcompensates.

Also, the diagnosed horizontal advection is relatively small and peaks during boreal spring (~May), very different from large boreal summer transport expected from the Asian summer monsoon (e.g. Ploeger et al 2012 explicitly calculate a 400 K maximum for water vapor horizontal mixing during August-October). Because of the fundamental problems in these important details, I am not convinced of the key result of large diagnosed vertical mixing from this analysis. As a note, it would have been helpful to have some statistical uncertainty estimates included with the results, although I did not recommend this in my previous review.

We think this is a misunderstanding on the reviewer's side. In fact, Fig. 1a in Ploeger et al. (2012) very clearly shows that the tendency due to horizontal mixing is largest during boreal spring: this tendency will be largest when the $H_2O$ difference between in-mixed extratropical air and tropical air is largest and this happens during boreal spring (e.g. difference between full red and black lines in Fig. 1A of Ploeger et al). The strongest effect in tropical mean tracer mixing ratio will be found when the accumulated tendency is strongest: this happens during late boreal summer when the tendency crosses through zero. The timing of this zero crossing is within a month between our result and Ploeger et al., so there is no discrepancy.

This has been further clarified in the manuscript.

The authors argue that the lack of strong vertical mixing derived using isentropic coordinates is physically consistent with the pressure coordinate calculations. However, I am not familiar with the quoted statement that "vertical mixing due to breaking gravity waves may be assumed to take place quasi-adiabatically", and a reference is not provided. The derived horizontal mixing for the isentropic case (Fig. 12) still peaks in the wrong season compared to previous work. Overall, the isentropic-coordinate calculations do not convince me that the pressure-coordinate calculations are correct (the p-coordinate calculations are more straightforward, and I believe they are not correct based on my comments above).

Basic gravity theory assumes adiabatic processes (e.g. Gill's book). In the stratosphere, diabatic processes are predominantly due to radiation. Radiative time-scales are long in the lowermost stratosphere (~30 days or longer) – long enough so that it seems reasonable to assume that gravity wave breaking happens fast enough to not "feel" any radiative damping. A clarifying statement has been included as footnote in the introduction.

We emphasize again that our isentropic coordinate results (based on the Mote et al control settings, so the associated tendencies are consistent with Mote et al) confirm existing results in the literature and that we feel that this provides some confidence in our approach and assumptions.

**Reviewer 2:**

We again thank the reviewer for her/his help to improve our paper. The reviewer suggests to run (and report on) more sensitivity tests for the seasonality in transport parameters. We have performed such tests and have now included additional text in the paper (section 3.2). In short, our score drops somewhat for reductions in the vertical advection seasonal cycle amplitude, slightly improves for a moderate enhancement (from 50% to 67%), but very strongly drops for more significant enhancements (beyond 75%). The score drops somewhat for very strong seasonality in K, but is otherwise hardly sensitive to changes in seasonality of the mixing parameters. In order to keep things simple we have opted to keep the 50% seasonality in all parameters.

**Reviewer 3 (Tim Dunkerton):**

We thank Tim Dunkerton for the overall encouraging remarks and the constructive criticism, which we address below. We have incorporated several corresponding changes in the manuscript.

This paper extends earlier analysis of the tape recorder signal in water vapor by including the seasonal cycle and focusing primarily on the lowermost part of the signal near 80 hPa. If I were interested in further research on this topic I would certainly want to consult this paper to build on the authors' results. I regard their findings as a partial improvement, perhaps half-way to where we need to go to understand fully the implications of water vapor behavior for mean vertical motion, vertical diffusion, and lateral in-mixing. The MLS data are inferior to HALOE for this purpose, the model is, well, just a model, and ERA-I is excessively diffusive, perhaps worthless of this purpose. I would probably rely on extending the HALOE analysis (Dunkerton, 2001 JAS) with the precision (for lowermost levels) illustrated by the authors.

We have now included several results based on HALOE data in the manuscript. Figures 3-6 now all include additional panels or lines based on HALOE. Versions of Fig. 2 and 7 based on HALOE are provided as supplement. The HALOE data set we used already came as zonal monthly means, which prevented us from running our analyses in isentropic coordinates (the first author now works at NCAR on a different project, so with the time constraint of a major revision we weren't able to run the entire isentropic coordinate analysis from instantaneous HALOE data, which is not as straightforward to process). Overall, results agree quite well between MLS and HALOE; if anything, HALOE suggests even greater vertical mixing strength. It is encouraging to see that the Mote et al. control setting performs much better for HALOE data (Fig. 6), on which it was based.

Two issues are overlooked that otherwise limit my appreciation for this work. First, it is now well established that the moist part of the tape recorder is driven by vertical, then lateral, transport of water vapor injected by overshooting convection in the Asian summer monsoon (Gettelman et al.,, 2004 JGR, et seq.). It is surprising if, in fact, no one has repeated Mote et al. with time-varying side boundary conditions to mimic this effect.

We agree that lateral transport, e.g. associated with the monsoons, may potentially be important, although Ploeger et al. (2012) conclude that it has a negligible effect on tropical water vapor just above the tropopause. The issue is that even though the monsoons may provide a great deal of mass transport into the TTL, the effect on tracers also depends on their background gradient. In the case of water vapor this background gradient is quite small during the relevant seasons.

Note that our 1-d model setup already included the effect of time-varying side boundary conditions: see text under Eq. 1 (section 3.2). This is now emphasized more in the text and the Gettelman et al. reference is included.

Second, although the authors seem to casually ascribe vertical diffusion to breaking gravity waves, it is well-known that such waves (if undergoing local convective instability in their phase of overturning) are not effective in mixing heat and constituents vertically (Coy et al., 1988 JAS). Inertia-gravity waves may undergo shear instability at large amplitude, altering this result possibly in a significant way (Dunkerton, 1985 JAS, et seq.).

We agree that our discussion of what may cause the vertical mixing is vague and hand-wavy. If gravity wave breaking contributes significantly then this has to happen in a non-classical way. The strong wave guide given by the large stratification jump and the tropopause inversion layer may perhaps give rise to non-linear effects that haven't been considered much in the past. Another possibility is that cloud-radiative effects modify diabatic damping in a way that gives rise to vertical dispersion, although this would then also be felt in isentropic coordinates. We admit that we don't have a clear conceptual picture at this point how this may work, but note that we are currently doing more work on this using cloud modeling.

Meanwhile, what is the role of overshooting convection in penetrating local theta surfaces? The main conclusion emerging from our work and subsequent studies of (de)hyrdration by overshooting convection is that ambient relative humidity determines the outcome: if high, leading to freeze-drying, if low, leading to hydration. While their latest additions to the text address dehydration, nothing is said to address hydration. In summary, while there are many reasons to publish this work, it will be a stepping stone, not close to any final answer, on a most intriguing problem. Publication is recommended.

We actually view hydration by overshooting convection as partially included in the vertical mixing term (technically the mixing is acting on total water in this case, which then becomes water vapor in the lowermost stratosphere through evaporation). We had mentioned this in the previous version (section 6, about half way through the paragraph discussing dehydration effects). Because of this we think that one shouldn't include an extra source term representing hydration in the 1-d model. But of course, an ad-hoc inclusion of a small source (e.g. due to hydration) during July-September together with a small sink (e.g. due to dehydration) during December-March would make our synthetic water vapor evolution agree with the observed one. Note however, that this is only true for MLS; our simulation already agrees very well with HALOE during July-September.

From a more philosophical point of view, one argument against inclusion of prescribed source / sink terms is the following. In the extreme case one could simulate the entire tape recorder purely by specifically designed sources and sinks, without any transport contributions – in strong conflict with what is well-known about lower stratospheric transport. In our view, it is preferable then to not include any prescribed source/sink term (as in Mote et al.) and rather diagnose and interpret missing tendencies after the fact.

Additional clarifying comments have been included in the text.

Minor suggestion: be sure to emphasize that only the lowermost part of the signal is analyzed. Whether your analysis benefits higher levels, with respect to annual & QBO influences on parameters, remains to be determined. I did some forward modeling prior to Mote et al. to ensure that vertical diffusivity must decrease rapidly with height above 80 hPa. Indeed, it must, otherwise the signal is too wide and decays much too fast. Frankly, if you have HALOE results not shown, I would add them prior to publication.

We feel that the text was already quite clear about the analyzed region: e.g. the title of the paper specifies that we study the "... tape recorder signal near the tropical tropopause". Qualifiers such as "in the lowermost stratosphere" or "just above the tropopause" are

frequently used. Nevertheless, we have now clearly stated in the abstract that the main analyzed level is 80 hPa.

Note that we use the same functional vertical structure for all transport parameters as in Mote et al., so the vertical decay above 80 hPa is implicitly included. Our variations in K represent a vertically uniform scaling factor onto the Mote et al. profile.

---

## Author Response (AR3)

**Response to Reviewer Comments, 3rd round of revisions**

We thank the reviewers for having another careful look at our manuscript. In particular, we are grateful to reviewer 2 who pointed out an inconsistency between Figures 5 & 6 regarding the new HALOE results we had included in the previous revision. It turns out that we had accidentally plotted the wrong "Mote et al" curve (red line in the bottom right panel in Fig. 6) – this has been corrected and the two Figures are now consistent.

For the current revision we upload the minor revision of the manuscript and this response; the abstract and supplement are unchanged from our previous version.